# DEFECTIVE CONVOLUTIONAL NETWORKS

## ABSTRACT

Robustness of convolutional neural networks (CNNs) has gained in importance on
account of adversarial examples, i.e., inputs added as well-designed perturbations
that are imperceptible to humans but can cause the model to predict incorrectly.
Recent research suggests that the noise in adversarial examples breaks the textural
structure, which eventually leads to wrong predictions. To mitigate the threat of
such adversarial attacks, we propose defective convolutional networks that make
predictions rely less on textural information but more on shape information by
properly integrating defective convolutional layers into standard CNNs. The de-
fective convolutional layers contain defective neurons whose activations are set to
be a constant function. As defective neurons contain no information and are far
different from standard neurons in its spatial neighborhood, the textural features
cannot be accurately extracted, and so the model has to seek other features for clas-
sification, such as the shape. We show extensive evidence to justify our proposal
and demonstrate that defective CNNs can defend against black-box attacks bet-
ter than standard CNNs. In particular, they achieve state-of-the-art performance
against transfer-based attacks without any adversarial training being applied.

## 1 INTRODUCTION

Deep learning (LeCun et al., 1998; 2015), especially deep Convolutional Neural Network
(CNN) (Krizhevsky et al., 2012), has led to state-of-the-art results spanning many machine learning
fields (Girshick, 2015; Chen et al., 2018; Luo et al., 2020). Despite the great success in numerous ap-
plications, recent studies show that deep CNNs are vulnerable to some well-designed input samples
named as Adversarial Examples (Szegedy et al., 2013; Biggio et al., 2013). Take the task of image
classification as an example, for almost every commonly used well-performed CNN, attackers are
able to construct a small perturbation on an input image, which is almost imperceptible to humans
but can make the model give a wrong prediction. The problem is serious as some well-designed ad-
versarial examples can be transferred among different kinds of CNN architectures (Papernot et al.,
2016b). As a result, a machine learning system can be easily attacked even if the attacker does not
have access to the model parameters, which seriously affect its use in practical applications.

There is a rapidly growing body of work on how to obtain a robust CNN, mainly based on adversarial
training (Szegedy et al., 2013; Goodfellow et al., 2015; Madry et al., 2017; Buckman et al., 2018;
Mao et al., 2019). However, those methods need lots of extra computation to obtain adversarial
examples at each time step and tend to overfit the attacking method used in training (Buckman et al.,
2018). In this paper, we tackle the problem in a perspective different from most existing methods.
In particular, we explore the possibility of designing new CNN architectures which can be trained
using standard optimization methods on standard benchmark datasets and can enjoy robustness by
themselves, without appealing to other techniques. Recent studies (Geirhos et al., 2017; 2018; Baker
et al., 2018; Brendel & Bethge, 2019) show that the predictions of standard CNNs mainly depend
on the texture of objects. However, the textural information has a high degree of redundancy and
may be easily injected with adversarial noise (Yang et al., 2019; Hosseini et al., 2019). Also,
Cao et al. (2020); Das et al. (2020) finds adversarial attack methods may perturb local patches to
contain textural features of incorrect classes. All the literature suggests that the wrong prediction
by CNNs for adversarial examples mainly comes from the change in the textural information. The
small perturbation of adversarial examples will change the textures and eventually affect the features
extracted by the CNNs. Therefore, a natural way to avoid adversarial examples is to let the CNN
make predictions relying less on textures but more on other information, such as the shape, which
cannot be severely distorted by small perturbations.

In practice, sometimes a camera might have mechanical failures which cause the output image to have many defective pixels (such pixels are always black in all images). Nonetheless, humans can still recognize objects in the image with defective pixels since we are able to classify the objects even in the absence of local textural information. Motivated by this, we introduce the concept of defectiveness into the convolutional neural networks: we call a neuron a defective neuron if its output value is fixed to zero no matter what input signal is received; similarly, a convolutional layer is a *defective convolutional layer* if it contains defective neurons. Before training, we replace the standard convolutional layers with the defective version on a standard CNN and train the network in the standard way. As defective neurons of the defective convolutional layer contain no information and are very different from their spatial neighbors, the textural information cannot be accurately extracted from the bottom defective layers to top layers. Therefore, we destroy local textural information to a certain extent and prompt the neural network to rely more on other information for classification. We call the architecture deployed with defective convolutional layers as *defective convolutional network*.

We find that applying the defective convolutional layers to the bottom[1] layers of the network and introducing various patterns for defective neurons arrangement across channels are critical. In summary, our main contributions are:

- We propose Defective CNNs and four empirical evidences to justify that, compared to standard CNNs, the defective ones rely less on textures and more on shapes of the inputs for making predictions.

- Experiments show that Defective CNNs has superior defense performance than standard CNNs against transfer-based attacks, decision-based attacks, and additive Gaussian noise.

- Using the standard training method, Defective CNN achieves state-of-the-art results against two transfer-based black-box attacks while maintaining high accuracy on clean test data.

- Through proper implementation, Defective CNNs can save a lot of computation and storage costs; thus may lead to a practical solution in the real world.

## 2   RELATED WORK

Various methods have been proposed to defend against adversarial examples. One line of research is to derive a meaningful optimization objective and optimize the model by adversarial training (Szegedy et al., 2013; Goodfellow et al., 2015; Huang et al., 2015; Madry et al., 2017; Buckman et al., 2018; Mao et al., 2019). The high-level idea of these works is that if we can predict the potential attack to the model during optimization, then we can give the attacked sample a correct signal and use it during training. Another line of research is to take an adjustment to the input image before letting it go through the deep neural network (Liao et al., 2017; Song et al., 2017; Samangouei et al., 2018; Sun et al., 2018; Xie et al., 2019; Yuan & He, 2020). The basic intuition behind this is that if we can clean the adversarial attack to a certain extent, then such attacks can be defended. Although these methods achieve some success, a major difficulty is that it needs a large extra cost to collect adversarial examples and hard to apply on large-scale datasets.

Several studies (Geirhos et al., 2017; 2018; Baker et al., 2018; Brendel & Bethge, 2019) show that the prediction of CNNs is mainly from the texture of objects but not the shape. Also, Cao et al. (2020); Das et al. (2020) found that adversarial examples usually perturb a patch of the original image to contain the textural feature of incorrect classes. For example, the adversarial example of the panda image is misclassified as a monkey because a patch of the panda skin is perturbed adversarially so that it alone looks like the face of a monkey (see Figure 11 in (Cao et al., 2020)). All previous works above suggest that the CNN learns textural information more than shape and the adversarial attack might come from textural-level perturbations. This is also correlated with robust features (Tsipras et al., 2018; Ilyas et al., 2019; Hosseini et al., 2019; Yang et al., 2019) which has attracted more interest recently. Pixels which encode textural information contain high redundancy and may be easily deteriorated to the distribution of incorrect classes. However, shape information is more compact and thus may serve as a more robust feature for predicting.

---

[1]In this paper, bottom layer means the layer close to the input and top layer means the layer close to the output prediction.

## 3 DEFECTIVE CONVOLUTIONAL NEURAL NETWORK

### 3.1 DESIGN OF DEFECTIVE CONVOLUTIONAL LAYERS

In this subsection, we introduce our proposed defective convolutional neural networks and discuss the differences between the proposed method and related topics.

First, we briefly introduce the notations. For one convolutional layer, denote $x$ as the input and $z$ as the output of neurons in the layer. Note that $x$ may be the input image or the output of the last convolutional layer. The input $x$ is usually a $M \times N \times K$ tensor in which $M/N$ are the height/width of a feature map, and $K$ is the number of feature maps, or equivalently, channels. Denote $w$ and $b$ as the parameters (e.g., the weights and biases) of the convolutional kernel. Then a standard convolutional layer can be mathematically defined as below.

**Standard convolutional layer:**

$$
\begin{aligned}
x' &= w * x + b, & (1) \\
z &= f(x'), & (2)
\end{aligned}
$$

where $f(\cdot)$ is a non-linear activation function such as ReLU[2] and $*$ is the convolutional operation. The convolutional filter receives signals in a patch and extracts local textural information from the patch. As mentioned in the introduction, recent works suggest that the prediction of standard CNNs strongly depends on such textural information, and noises imposed on the texture may lead to wrong predictions. Therefore, we hope to learn a feature extractor which does not solely rely on textural features and also considers other information. To achieve this goal, we introduce the *defective convolutional layer* in which some neurons are purposely designed to be corrupted. Define $M_{\text{defect}}$ to be a binary matrix of size $M \times N \times K$. Our defective convolutional layer is defined as follows.

**Defective convolutional layer:**

$$
\begin{aligned}
x' &= w * x + b, & (3) \\
z' &= f(x') & (4) \\
z &= M_{\text{defect}} \circ z', & (5)
\end{aligned}
$$

where $\circ$ denotes element-wise product. $M_{\text{defect}}$ is a fixed matrix and is not learnable during training and testing. We can see that $M_{\text{defect}}$ plays a role of "masking" out values of some neurons in the layer. This disturbs the distribution of local textural information and decouples the correlation among neurons. With the masked output $z$ as input, the feature extractor of the next convolutional layer cannot accurately capture the local textural feature from $x$. As a consequence, the textural information is hard to pass through the defective CNN from bottom to top. To produce accurate predictions, the deep neural network has to find relevant signals other than the texture, e.g., the shape. Those corrupted neurons have no severe impact on the extraction of shape information since neighbors of those neurons in the same filter are still capable of passing the shape information to the next layer.

In this paper, we find that simply setting $M_{\text{defect}}$ by random initialization is already helpful for learning a robust CNN. Before training, we sample each entry in $M_{\text{defect}}$ using Bernoulli distribution with keep probability $p$ and then fix $M_{\text{defect}}$ during training and testing. More discussions and ablation studies are provided in Section 4.

As can be seen from Equation (3)-(5), the implementation of our defective convolutional layer is similar to the dropout operation (Srivastava et al., 2014). To demonstrate the relationship and differences, we mathematically define the dropout as below.

**Standard convolutional layer + dropout:**

$$
\begin{aligned}
M_{\text{dropout}} &\sim \text{Bernoulli}(p) & (6) \\
x' &= w * x + b & (7) \\
z' &= f(x') & (8) \\
z &= M_{\text{dropout}} \circ z'. & (9)
\end{aligned}
$$

---

[2]Batch normalization is popularly used on $x'$ before computing $z$. Here we simply omit this.

The shape of $M_{\text{dropout}}$ is the same as $M_{\text{defect}}$, and the value of each entry in $M_{\text{dropout}}$ is sampled in each batch using some sampling strategies at each step during training. Generally, entries in $M_{\text{dropout}}$ are independent and identically sampled in an online fashion using Bernoulli distribution with keep probability $p$.

There are several significant differences between dropout and defective convolutional layer. First, the binary matrix $M_{\text{dropout}}$ is sampled online during training and is removed during testing, while the binary matrix $M_{\text{defect}}$ in defective convolutional layers is predefined and keeps fixed in both training and testing. The predefined way can help Defective CNNs save a lot of computation and storage costs. Second, the motivations behind the two methods are quite different and may lead to differences in the places to applying methods, the values of the keep probability $p$, and the shape of the masked unit. Dropout tries to reduce overfitting by preventing co-adaptations on training data. When comes to CNNs, those methods are applied to top layers, $p$ is set to be large (e.g., $0.9$), and the masked units are chosen to be a whole channel in Tompson et al. (2015) and a connected block in Ghiasi et al. (2018). However, our method tries to prevent the model extract textural information of inputs for making predictions. We would apply the method to bottom layers, use a small $p$ (e.g. $0.1$), and the masked unit is a single neuron. Also, in our experiments, we will show that the proposed method can improve the robustness of CNNs against transfer-based attacks and decision-based attacks, while the dropout methods cannot.

## 3.2 Defective CNNs Rely Less on Texture but More on Shape for Predicting

In this subsection, we provide extensive analysis to show Defective CNNs that, compared to the standard CNNs, rely less on textures and more on shapes of the inputs for making predictions.

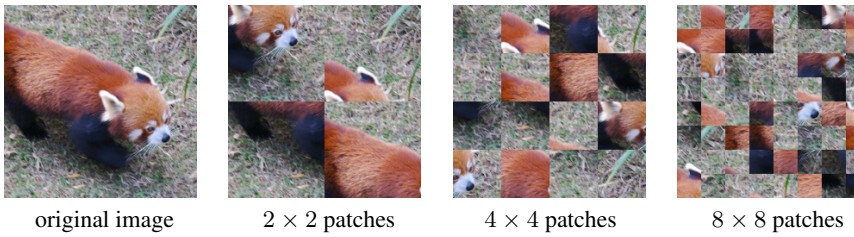

original image        $2 \times 2$ patches        $4 \times 4$ patches        $8 \times 8$ patches

Figure 1: An example image that is randomly shuffled after being divided into $2 \times 2$, $4 \times 4$ and $8 \times 8$ patches respectively.

First, we design a particular image manipulation in which the local texture of the object in an image is preserved while the shape is destroyed. Particularly, we divide an image into $k \times k$ patches and randomly relocate those patches to form a new image. An example is shown in Figure 1. A model that more focuses on the shape cues should achieve lower performance on such images while it is trained on the normal dataset. We manipulate a set of images and test whether a defective CNN and a standard CNN can make correct predictions. The experimental details are described as follows. We first construct a defective CNN by applying defective convolutional layers to the bottom layers of a standard ResNet-18, and train the defective CNN along with a standard ResNet-18 on the ImageNet dataset. Then, we sample images from the validation set which are predicted correctly by both of the CNNs. We make manipulations to the sampled images by setting $k \in \{2, 4, 8\}$, feed these images to the networks and check their classification accuracy. The results in Table 1, 13 show that when the shape information is destroyed but the local textural information is preserved, Defective CNNs perform consistently worse than standard CNNs, thus verifying out intuition.

| Model | $2 \times 2$ | $4 \times 4$ | $8 \times 8$ | IN $\to$ SIN |
|---|---|---|---|---|
| Standard CNN | 99.53% | 84.36% | 20.08% | 15.33% |
| Defective CNN | 96.32% | 56.91% | 9.04% | 20.20% |

Table 1: Left three columns are the accuracy of classifying randomly shuffled images. The rightmost column is the accuracy of training on ImageNet and testing on Stylized-ImageNet. The phenomena are similar for different architectures and can be found in Appendix A.8.

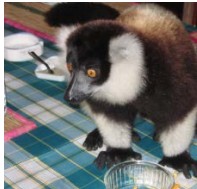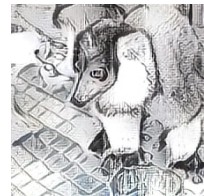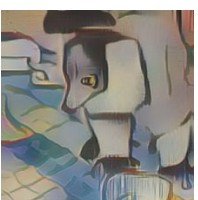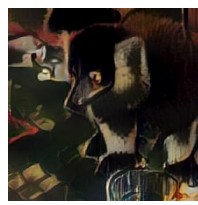

Figure 2: The leftmost is an image in the ImageNet, the right three are the corresponding images in the Stylized-ImageNet.

Second, we test on the Stylized-ImageNet (Geirhos et al., 2018), a stylized version of ImageNet, where the local textures of images are changed, while global object shapes remain (See examples in Figure 2). A model that more focuses on the shape cues should achieve higher performance on the Stylized-ImageNet while it is trained on the ImageNet. We test on the same model used in the randomly shuffled experiments by feeding the images from the validation set of Stylized-ImageNet whose corresponding images in ImageNet can be correctly classified by both two tested models. The result in Table 1, 13 show that Defective CNNs achieves consistently higher transferring accuracy than standard CNNs, thus verifying our argument.

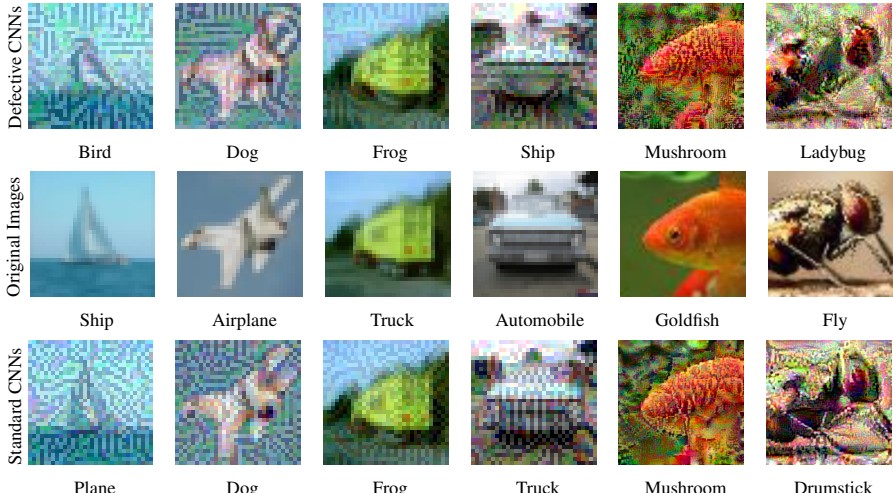

Figure 3: **First row**: the adversarial examples and the labels predicted by Defective CNNs. **Second row**: the original images and the ground truth labels. **Third row**: the adversarial examples and the labels predicted by standard CNNs. Attack method is MIFGSM (Dong et al., 2017) and the perturbation scales are $\ell_\infty \in \{16/255, 32/255\}$. More details can be found in Appendix B.

From another perspective, if a model makes predictions relying more on shape information, the manipulation of the shape of objects will play a larger role in generating adversarial examples. To verify this, we train defective and standard CNNs on CIFAR-10 and Tiny-ImageNet, and then attack on the validation set. Figure 3 shows some examples. We can see that adversarial examples against the defective CNNs change the shape of the objects and may even fool humans as well. Compare with the adversarial examples of Figure 9 in Qin et al. (2020), our adversarial examples exhibit more salient characteristics of the adversarial classes. Also, we conduct a user study in Appendix B to show that the adversarial examples generated by Defective CNNs, compared to the standard ones, are more perceptually like the adversarial classes. The phenomenon not only supports our intuition, but also is consistent with the findings in Tsipras et al. (2018); Qin et al. (2020) that the representations learned by robust models tend to align better with human perception.

Furthermore, perturbations generated by standard CNNs and additive Gaussian noises usually would not affect the shape information (Szegedy et al., 2013; Ford et al., 2019). A model is supposed to recognize those adversarial examples better if it relies much on shape information for predictions. In Section 4, we show that defective CNNs achieve higher defense performance than standard CNN against the two types of attack.

## 4 EXPERIMENTS

In real-world tasks, attackers usually cannot access the parameters of the target models and thus need to transfer adversarial examples generated by their models. This setting of attack is referred to as transfer-based attacks (Liu et al., 2016; Kurakin et al., 2016). Sometimes, attackers can get the final model decision and raise the more powerful decision-based attacks (Brendel et al., 2017). Both the two types of black-box attack are available in most real-world scenarios and should be considered. Recently, Ford et al. (2019) bridge the adversarial robustness and corruption robustness (Hendrycks & Dietterich, 2018), and points out that a successful adversarial defend method should also effectively defend against additive Gaussian noise. Therefore, to meet the requirements for practical systems, we examine the performance of models against transfer-based attacks, decision-based attacks, and additive Gaussian noise.

In the following sections, we evaluate the robustness of defective CNNs with different architectures and compare with state-of-the-art defense methods against transfer-based attacks, and then make ablation studies on possible design choices of defective CNN. Due to space limitation, we list the experiments of decision-based attacks, additive Gaussian noise, gray-box attacks, white-box attacks, and more results of transfer-based attacks in Appendix A.

Note that, in this paper, all the successful defense rates except the rates listed in Table 2, 3 are calculated on the samples whose corresponding original images can be classified correctly by the tested model. This can erase the influence of test accuracy that different models have different test accuracy on clean data, and thus help evaluate the robustness of models.

### 4.1 TRANSFER-BASED ATTACK

#### 4.1.1 EXPERIMENTAL SETTINGS

We evaluate the defense performance of Defective CNNs against transfer-based attacks and compare with state-of-the-art defense methods on CIFAR-10 and Tiny-ImageNet. For CIFAR-10, we follow the setting used in Madry et al. (2017), and use a standard ResNet-18 to generate adversarial examples by FGSM (Goodfellow et al., 2015) and PGD (Kurakin et al., 2016). The two attack methods both have perturbation scale $\ell_\infty = 8/255$ and PGD runs for 7 gradient descent steps with step size $2/255$. For Tiny-ImageNet, we follow the setting used in Mao et al. (2019) and use a standard ResNet-50 to generate adversarial examples by PGD with $\ell_\infty = 8/255$, steps 20, and step size $2/255$. We would compare with two types of defense methods including the variants of adversarial training (Madry et al., 2017; Kannan et al., 2018; Mao et al., 2019) and approaches that try to erase the adversarial noise of inputs (Wang & Yu, 2019; Addepalli et al., 2020; Yuan & He, 2020). To validate the difference between the proposed method and dropout methods, we also compare with two CNN variants SpatialDropout (Tompson et al., 2015) and DropBlock (Ghiasi et al., 2018). For both methods, we follow the instruction from Ghiasi et al. (2018) to apply dropout to the $\{3^{\text{rd}}, 4^{\text{th}}\}$ block with keep probability $p = 0.9$. The block of DropBlock is set to be a $3 \times 3$ square. For our method, we use the corresponding network structure but applying defective convolutional layers to the bottom layers (see illustrations in Appendix C). We use keep probability $p = 0.1$ and train the model with the standard optimization method. Training details and curves can be found in Appendix D.

Second, we test our proposed method in different architectures on the CIFAR-10 dataset. We apply defective convolutional layers, in a way which is similar to the experiment above, to five popular network architectures: ResNet-18 (He et al., 2016), ResNet-50, DenseNet-121 (Huang et al., 2017), SENet-18 (Hu et al., 2017b) and VGG-19 (Simonyan & Zisserman, 2014). For each architecture, we replace the standard convolutional layer with the defective version on the bottom layers (see illustrations in Appendix C). We then test the black-box defense performance against transfer-based attacks on 5000 samples from the validation set. Adversarial examples are generated by PGD, which runs for 20 steps with step size 1 and the $\ell_\infty$ perturbation scale is set to $16/255$. Results on MNIST can be found in Appendix A.

#### 4.1.2 EXPERIMENTAL RESULTS

Table 2, 3 show the results on CIFAR-10 and Tiny-ImageNet, respectively. We can see the proposed method outperforms all the adversarial training variants which need extra training costs and most of the cleaning inputs methods. Although Yuan & He (2020) is competitive with the proposed method

in CIFAR-10, it need to collect adversarial examples and run inner loops, thus largely increase times-tamps. Also, we can conclude that spatial dropout and drop block do not improve the robustness of standard CNNs. The results show the strengths of our proposed method on both robustness and generalization, even though our model is only trained on *clean data*. Also, it is interesting that the CNNs can maintain such clean accuracy even 90% neurons in bottom layers are dropped.

| Method | FGSM | PGD | Clean Acc |
|---|---|---|---|
| Standard CNN | 55.92% | 15.96% | 95.03% |
| Standard CNN + SD (Tompson et al., 2015) | 52.11% | 12.98% | 95.44% |
| Standard CNN + DB (Ghiasi et al., 2018) | 56.27% | 14.69% | 95.38% |
| BPFC (Addepalli et al., 2020) | 75.52% | 77.07% | 82.30% |
| Adv. Training (Madry et al., 2017) | 77.10% | 78.10% | 87.14% |
| TLA (Mao et al., 2019) | - | 83.20% | 86.21% |
| Adv. Network (Wang & Yu, 2019) | 77.23% | 74.04% | 91.32% |
| EGC-FL (Yuan & He, 2020) | 79.09% | 82.78% | 91.65% |
| Defective CNN | 77.93% | 84.60% | 91.44% |

Table 2: Defense performance on CIFAR-10.

| Method | PGD | Clean Acc |
|---|---|---|
| Standard CNN | 9.99% | 60.64% |
| Standard CNN + SD (Tompson et al., 2015) | 8.43% | 61.82% |
| Standard CNN + DB (Ghiasi et al., 2018) | 9.15% | 61.37% |
| Adv. Training (Madry et al., 2017) | 27.73% | 44.77% |
| ALP (Kannan et al., 2018) | 30.31% | 41.53% |
| TLA (Mao et al., 2019) | 29.98% | 40.89% |
| Defective CNN | 32.32% | 55.74% |

Table 3: Defense performance on Tiny-ImageNet.

Second, we list the black-box defense results of applying defective convolutional layers to various architectures in Table 4. The results show that defective convolutional layers consistently improve the robustness of various network architectures against transfer-based attacks. We can also see that the trend of robustness increases as the keep probability becomes smaller.

| Architecture | ResNet-18 | ResNet-50 | DenseNet-121 | SENet-18 | VGG-19 | Test Accuracy |
|---|---|---|---|---|---|---|
| ResNet-18 | 5.98% | 0.94% | 14.14% | 3.32% | 26.97% | 95.33% |
| 0.5-Bottom | 53.89% | 33.05% | 70.38% | 57.52% | 58.66% | 93.39% |
| 0.3-Bottom | 78.23% | 67.64% | 86.99% | 82.46% | 77.57% | 91.83% |
| ResNet-50 | 16.61% | 0.22% | 14.60% | 12.26% | 42.38% | 95.25% |
| 0.5-Bottom | 51.55% | 17.61% | 62.69% | 53.82% | 62.73% | 94.43% |
| 0.3-Bottom | 71.63% | 48.03% | 80.94% | 75.91% | 75.72% | 93.46% |
| DenseNet-121 | 14.53% | 0.60% | 2.98% | 7.79% | 31.57% | 95.53% |
| 0.5-Bottom | 35.07% | 8.01% | 34.21% | 30.86% | 45.28% | 94.34% |
| 0.3-Bottom | 58.19% | 33.86% | 62.32% | 59.74% | 62.09% | 92.82% |
| SENet-18 | 6.72% | 0.90% | 12.29% | 2.23% | 26.86% | 95.09% |
| 0.5-Bottom | 52.95% | 30.78% | 66.81% | 52.49% | 57.45% | 93.53% |
| 0.3-Bottom | 74.73% | 59.42% | 84.31% | 78.72% | 75.04% | 92.54% |
| VGG-19 | 33.46% | 14.16% | 49.76% | 29.98% | 21.20% | 93.93% |
| 0.5-Bottom | 72.27% | 59.70% | 83.50% | 77.93% | 66.75% | 91.73% |
| 0.3-Bottom | 85.53% | 79.20% | 91.01% | 88.51% | 81.92% | 90.11% |

Table 4: Black-box defense performances on CIFAR-10. Networks in the first row are the source models for generating adversarial examples by PGD. 0.5-Bottom and 0.3-Bottom mean applying defective convolutional layers with keep probability $0.5$ and $0.3$ to the bottom layers of the network whose name lies just above them. The source and target networks are initialized differently if they share the same architecture. Numbers in the middle mean the success defense rates.

## 4.2 ABLATION STUDIES

There are several design choices of the defective CNN, which include the appropriate positions to apply defective convolutional layers, the choice of the keep probabilities, the benefit of breaking symmetry, as well as the diversity introduced by randomness. In this subsection, we conduct a series of comparative experiments and use black-box defense performance against transfer-based attacks as the evaluation criterion. In our experiments, we found that the performance is not sensitive to the choices on the source model to attack and the target model to defense. Here, we only list the performances using DenseNet-121 as the source model and ResNet-18 as the target model on the CIFAR-10 dataset and leave more experimental results in Appendix A.10. The results are listed in Table 5.

**Defective Layers on Bottom layers vs. Top Layers, Keep Probabilities.** We apply defective layers with different keep probabilities to the bottom layers and the top layers of the standard CNNs (see illustrations in Appendix C). Comparing the results of the models with the same keep probability but different parts being masked, we find that applying defective layers to *bottom* layers enjoys significantly higher success defense rates, while applying to the top layers cannot. This corroborates the phenomena shown in literature (Zeiler & Fergus, 2014; Mordvintsev et al., 2015), where bottom

| Architecture | $\text{FGSM}_{16}$ | $\text{PGD}_{16}$ | $\text{PGD}_{32}$ | $\text{CW}_{40}$ | Test Accuracy |
|---|---|---|---|---|---|
| Standard CNN | 14.91% | 14.14% | 7.16% | 8.23% | 95.33% |
| 0.7-Bottom | 23.29% | 51.29% | 37.00% | 36.95% | 94.03% |
| 0.5-Bottom | 30.86% | 70.38% | 56.36% | 54.02% | 93.39% |
| 0.3-Bottom | 48.57% | 86.99% | 78.41% | 73.70% | 91.83% |
| 0.7-Top | 14.62% | 10.55% | 4.91% | 7.88% | 95.16% |
| 0.5-Top | 10.76% | 11.06% | 5.10% | 7.19% | 94.94% |
| 0.3-Top | 11.23% | 11.77% | 5.80% | 10.10% | 94.61% |
| 0.7-Bottom, 0.7-Top | 24.15% | 45.12% | 30.24% | 29.65% | 94.16% |
| 0.7-Bottom, 0.3-Top | 11.26% | 33.67% | 20.28% | 23.31% | 93.44% |
| 0.3-Bottom, 0.3-Top | 27.43% | 75.49% | 62.78% | 62.47% | 89.78% |
| 0.3-Bottom, 0.7-Top | 40.58% | 82.77% | 72.15% | 68.58% | 91.23% |
| 0.5-Bottom | 30.86% | 70.38% | 56.36% | 54.02% | 93.39% |
| 0.1-Bottom | 79.93% | 96.70% | 94.68% | 89.67% | 87.68% |
| $0.5\text{-Bottom}_{\text{DC}}$ | 12.15% | 19.93% | 11.20% | 12.72% | 95.12% |
| $0.1\text{-Bottom}_{\text{DC}}$ | 19.00% | 53.87% | 41.40% | 44.80% | 93.27% |
| $0.5\text{-Bottom}_{\text{SM}}$ | 48.86% | 85.00% | 75.60% | 72.07% | 92.57% |
| $0.1\text{-Bottom}_{\text{SM}}$ | 39.40% | 80.36% | 72.43% | 65.38% | 74.28% |

Table 5: Ablation experiments of defective CNNs. $p$-**Bottom** and $p$-**Top** mean applying defective layers with keep probability $p$ to bottom layers and top layers respectively. $p$-**Bottom**$_{\text{DC}}$ means making whole channels defective with keep probability $p$. $p$-**Bottom**$_{\text{SM}}$ means using the same defective mask in every channel with keep probability $p$. $\text{FGSM}_{16}$, $\text{PGD}_{16}$ and $\text{PGD}_{32}$ denote attack method FGSM with perturbation scale $\ell_\infty = 16/255$, PGD with perturbation scale $\ell_\infty = 16/255$ and $32/255$ respectively. $\text{CW}_{40}$ denotes CW attack method (Carlini & Wagner, 2016) with confidence $\kappa = 40$. Numbers in the middle mean the success defense rates.

layers mainly contribute to detect the edges and shape, while the receptive fields of neurons in top layers are too large to respond to the location sensitive information. Also, we find that the defense accuracy monotonically increases as the test accuracy decreases along with the keep probability (See the trend map in Appendix A.9). The appropriate value for the keep probability mainly depends on the relative importance of generalization and robustness. Another practical way is to ensemble Defective CNNs with different keep probabilities.

**Defective Neuron vs. Defective Channel.** As our method independently selects defective neurons on different channels in a layer, we break the symmetry of the original CNN structure. To see whether this asymmetric structure would help, we try to directly mask whole channels instead of neurons using the same keep probability as the defective layer and train it to see the performance. This defective channel method does not hurt the symmetry while also leading to the same decrease in the number of convolutional operations. Table 5 shows that although our defective CNN suffers a small drop in test accuracy due to the low keep probability, we have a great gain in the robustness, compared with the defective-channel CNN.

**Defective Masks are Shared Among Channels or Not.** The randomness in generating masks in different channels and layers allows each convolutional filter to focus on different input patterns. Also, it naturally involves various topological structures for local feature extraction instead of the expensive learning way (Dai et al., 2017; Chang et al., 2018; Zhu et al., 2019). We show the essentiality of generating various masks per layer via experiments that compare to a method that only randomly generates one mask per layer and uses it in every channel. Table 5 shows that applying the same mask to each channel will decrease the test accuracy. This may result from the limitation of expressivity due to the monotone masks at every channel of the defective layer.

## 5 CONCLUSION

In this paper, we introduce and experiment on defective CNNs, a modified version of existing CNNs that makes CNNs capture more information other than local textures, especially the shape. We propose four empirical evidence to justify this and also show that Defective CNNs can achieve high robustness against black-box attacks while maintaining high test accuracy. Another insight resulting from our experiments is that the adversarial perturbations generated against defective CNNs can actually change the semantic information of images and may even "fool" humans. We hope that these findings bring more understanding on adversarial examples and the robustness of neural networks.

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

DEFECTIVE CONVOLUTIONAL NETWORKS APPENDIX

TABLE OF CONTENTS

## A    MORE EXPERIMENTAL RESULTS

### A.1    DECISION-BASED ATTACK

#### A.1.1    EXPERIMENTAL SETTINGS

In this subsection, we evaluate the defense performance of networks with defective convolutional layers against the decision-based attack. Decision-based attack performs based on the prediction of the model. It needs less information from the model and has the potential to perform better against adversarial defenses based on gradient masking. Boundary attack (Brendel et al., 2017) is one effective decision-based attack. The attack will start from a point that is already adversarial by applying a large scale perturbation to the original image and keep decreasing the distance between the original image and the adversarial example by random walks. After iterations, we will get the final perturbation, which has a relatively small scale. The more robust the model is, the larger the final perturbation will be.

In our experiments, we use the implementation of boundary attack in Foolbox (Rauber et al., 2017). It finds the adversarial initialization by simply adding large scale uniform noise on input images. We perform our method on ResNet-18 and test the performance on CIFAR-10 with 500 samples from the validation set. The 5-block structure of ResNet-18 is shown in Appendix Figure 2. The blocks are labeled $0, 1, 2, 3, 4$ and the $0^{th}$ block is the first convolution layer. We apply the defective layer structure with keep probability $p = 0.1$ to the bottom blocks (the $0^{th}$, $1^{st}$, $2^{nd}$ blocks). For comparison, we implement label smoothing (Szegedy et al., 2016) with smoothing parameter $\epsilon = 0.1$ on a standard ResNet-18, and spatial dropout and drop block with the setting same as Section 4.1.1.

#### A.1.2    EXPERIMENTAL RESULTS

We use the median squared to evaluate the performance, which is defined as $\ell_2$-distance of final perturbation across all samples proposed in (Brendel et al., 2017). The score $S(M)$ is defined as $Median_i \left( \frac{1}{N} \|P_i^M\|_2^2 \right)$, where $P_i^M \in \mathbb{R}^N$ is the final perturbation that the Boundary attack finds on model $M$ for the $i^{th}$ image. Before computing $P_i^M$, the images are normalized into $[0, 1]^N$.

| Model | $S(M)$ |
|---|---|
| Standard CNN | 7.3e-06 |
| Standard CNN + SD (Tompson et al., 2015) | 7.2e-06 |
| Standard CNN + DB (Ghiasi et al., 2018) | 6.1e-06 |
| Standard CNN + LS (Szegedy et al., 2016) | 6.8e-06 |
| Defective CNN | **3.5e-05** |

Table 6: Black-box defense performances against decision-based attack. $S(M)$ is defined above. The larger value $S(M)$ has, the more robust the model is. SD, DB, and LS denote spatial dropout, drop block, and label smoothing, respectively. Detailed settings are listed in Section A.1.

From the results in Table 6, we point out that spatial dropout and drop block can not enhance the robustness against the boundary attack. Neither does the label smoothing technique. This is consistent with the discovery in Section 4.1.1, and in Papernot et al. (2016a) where they point out that label smoothing is a kind of gradient masking method. Also, the defective CNN achieves higher performance over the standard CNN.

### A.2    ADDITIVE GAUSSIAN NOISE

#### A.2.1    EXPERIMENTAL SETTINGS

In this subsection, we evaluate the defense performance of networks with defective convolutional layers against additive Gaussian noise. Recently, Ford et al. (2019) bridge the adversarial robustness and corruption robustness, and points out that a successful adversarial defense method should also effectively defend against additive Gaussian noise. Also the Gaussian noises usually do not change the shape of objects, our models should have better defense performance. To see whether our structure is more robust in this setting, we feed input images with additive Gaussian noises to both standard and defective CNNs.

To obtain noise of scales similar to the adversarial perturbations, we generate i.i.d. Gaussian random variables $x \sim N(0, \sigma^2)$, where $\sigma \in \{1, 2, 4, 8, 12, 16, 20, 24, 28, 32\}$, clip them to the range $[-2\sigma, 2\sigma]$ and then add them to every pixel of the input image. Note that, the magnitude range of Gaussian noises used in our experiments covers all 5 severity levels used in Hendrycks & Dietterich (2018). For CIFAR-10, we add Gaussian noises to 5000 samples which are drawn randomly from the validation set and can be classified correctly by all the tested models. We place the defective layers with keep probability $p = 0.1$ on ResNet-18 in the same way as we did in Section A.1.

### A.2.2 EXPERIMENTAL RESULTS

The experimental results are shown in Figure 4. The standard CNN is still robust to small scale Gaussian noise such as $\sigma \leq 8$. After that, the performance of the standard CNN begins to drop sharply as $\sigma$ increases. In contrast, defective CNNs show far better robustness than the standard version. The defective CNN with keep probability $0.1$ can maintain high accuracy until $\sigma$ increase to 16 and have a much slower downward trend as $\sigma$ increases.

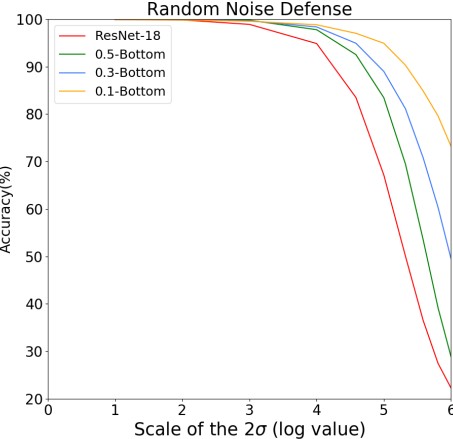

Figure 4: Defense performance against additive Gaussian noise. $p$-Bottom means applying defective convolutional layers with keep probability $p$ to the bottom layers of a standard ResNet-18.

### A.3 TRANSFER-BASED ATTACK ON WIDE-RESNET

In this subsection, we also evaluate the defense performance of networks with defective convolutional layers against transfer-based attacks. We compare our proposed method with two adversarial training methods (Buckman et al., 2018; Madry et al., 2017). For fair comparisons, we follow Buckman et al. (2018) to generate adversarial examples using wide residual networks (Zagoruyko & Komodakis, 2016) with a depth of 32 and a width factor of 4. The 4-block structure of ResNet-32 is shown in Appendix C. The blocks are labeled $0, 1, 2, 3$ and the $0^{\text{th}}$ block is the first convolution layer. Both FGSM (Goodfellow et al., 2015) and PGD (Kurakin et al., 2016) attacks are run on the entire validation set of CIFAR-10 dataset. These two methods both have $\ell_\infty$ perturbation scale $8/255$ and PGD runs for 7 gradient descent steps with step size 2. The generated adversarial examples are used to attack target networks. For the target network, we use the same structure but applying defective convolutional layers to the $0^{\text{th}}$ and $1^{\text{st}}$ blocks with keep probability $p = 0.1$ and train the model using the standard optimization method. As is mentioned in Section 3.1, our proposed method is essentially different from dropout, and thus we also take dropout methods as baselines. More specifically, we test SpatialDropout and DropBlock. For both methods, we follow the instruction from Ghiasi et al. (2018) to apply dropout to the $3^{\text{rd}}$ block with $p = 0.9$. The block of DropBlock is set to be a $3 \times 3$ square. The result is listed in Table 7.

| Model | FGSM | PGD | Test Accuracy |
|---|---|---|---|
| Standard CNN | 52.88% | 15.98% | 95.39% |
| Standard CNN + SD (Tompson et al., 2015) | 51.63% | 14.28% | 95.98% |
| Standard CNN + DB (Ghiasi et al., 2018) | 51.55% | 12.71% | 95.81% |
| Adversarial Training (Madry et al., 2017) | 85.60% | 86.00% | 87.30% |
| Thermometer(16) (Buckman et al., 2018) | - | 88.25% | 89.88% |
| Thermometer(32) (Buckman et al., 2018) | - | 86.60% | 90.30% |
| Defective CNN | **86.24%** | **88.43%** | **91.12%** |

Table 7: Black-box defense performances against transfer-based attacks. SD and DB denote spatial dropout and drop block, respectively.

## A.4 TRANSFER-BASED ATTACK FROM ENSEMBLE MODELS ON CIFAR-10

In this subsection, we evaluate the defense performance of networks with defective convolutional layers against transfer-based attack from ensemble models on the CIFAR-10 dataset. We apply defective convolutional layers to five popular network architectures ResNet-18, ResNet-50 (He et al., 2016), DenseNet-121, SENet-18 (Hu et al., 2017a), VGG-19 (Simonyan & Zisserman, 2014), and test the black-box defense performance against transfer-based attacks from ensemble models on the CIFAR-10 dataset. For each architecture, we replace the standard convolutional layer with the defective version on the bottom layers of different architectures. Illustrations of defective layers applied to these network architectures can be found in Appendix C. We test the black-box defense performance against transfer-based attacks on 5000 samples from the validation set. Adversarial examples are generated by PGD, which runs for 7 steps with step size 2 and the $\ell_\infty$ perturbation scale is set to $8/255$. We generate five ensemble models as the source model by fusing every four models in all five models.

The results can be found in Table 8. These results show that defective convolutional layers can consistently improve the black-box defense performance of various network architectures against transfer-based attacks from ensemble models on the CIFAR-10 dataset.

| Architecture | –ResNet-18 | –ResNet-50 | –DenseNet-121 | –SENet-18 | –VGG-19 | Test Accuracy |
|---|---|---|---|---|---|---|
| ResNet-18 | 1.02% | 0.74% | 0.76% | 0.94% | 0.88% | 95.33% |
| 0.5-Bottom | 32.98% | 35.95% | 29.36% | 32.31% | 37.24% | 93.39% |
| 0.3-Bottom | 69.52% | 72.44% | 67.02% | 68.63% | 72.23% | 91.83% |
| ResNet-50 | 1.07% | 2.32% | 1.31% | 1.17% | 0.82% | 95.25% |
| 0.5-Bottom | 23.61% | 31.52% | 21.20% | 22.89% | 25.59% | 94.43% |
| 0.3-Bottom | 55.47% | 62.43% | 53.25% | 55.13% | 58.47% | 93.46% |
| DenseNet-121 | 0.70% | 0.88% | 1.37% | 0.74% | 0.58% | 95.53% |
| 0.5-Bottom | 6.99% | 10.19% | 7.77% | 6.93% | 8.10% | 94.34% |
| 0.3-Bottom | 32.07% | 38.28% | 31.59% | 31.01% | 35.42% | 92.82% |
| SENet-18 | 0.91% | 0.66% | 0.74% | 0.92% | 0.64% | 95.09% |
| 0.5-Bottom | 29.16% | 32.58% | 25.84% | 29.17% | 32.88% | 93.53% |
| 0.3-Bottom | 62.12% | 65.83% | 59.94% | 62.90% | 65.64% | 92.54% |
| VGG-19 | 8.27% | 8.28% | 6.42% | 7.64% | 14.08% | 93.93% |
| 0.5-Bottom | 60.62% | 63.71% | 57.85% | 59.28% | 64.73% | 91.73% |
| 0.3-Bottom | 80.97% | 82.84% | 80.06% | 80.04% | 82.86% | 90.11% |

Table 8: Black-box defense performances against transfer-based attacks from ensemble models on the CIFAR-10 dataset. Numbers in the middle mean the success defense rates. Networks in the first row indicate the source models which ensemble other four models except for the network itself. The source model generates adversarial examples by PGD. 0.5-Bottom and 0.3-Bottom mean applying defective convolutional layers with keep probability 0.5 and 0.3 to the bottom layers of the network whose name lies just above them. The source and target networks are initialized differently if they share the same architecture.

A.5    TRANSFER-BASED ATTACK ON MNIST

In this subsection, we evaluate the defense performance of networks with defective convolutional layers against trasfer-based attack on the MINST dataset. We apply defective convolutional layers to five popular network architectures ResNet-18, ResNet-50, DenseNet-121, SENet-18, VGG-19, and test the black-box defense performance against transfer-based attacks on MNIST dataset. For each architecture, we replace the standard convolutional layer with the defective version on bottom layers of different architectures. Illustrations of defective layers applied to these network architectures can be found in Appendix C. We test the black-box defense performance against transfer-based attacks on 5000 samples from the validation set. Adversarial examples are generated by PGD which runs for 40 steps with step size $0.01 \times 255$ and perturbation scale $0.3 \times 255$.

The results can be found in Table 9. These results show that defective convolutional layers can consistently improve the black-box defense performance of various network architectures against transfer-based attacks on the MNIST dataset.

| Architecture | ResNet-18 | ResNet-50 | DenseNet-121 | SENet-18 | VGG-19 | Test Accuracy |
|---|---|---|---|---|---|---|
| ResNet-18 | 0.06% | 24.13% | 1.66% | 0.14% | 9.57% | 99.49% |
| 0.5-Bottom | 3.49% | 43.04% | 8.66% | 7.66% | 26.47% | 99.34% |
| 0.3-Bottom | 25.91% | 75.59% | 36.19% | 38.03% | 64.63% | 99.29% |
| ResNet-50 | 2.93% | 9.30% | 7.68% | 5.94% | 19.68% | 99.39% |
| 0.5-Bottom | 8.54% | 23.06% | 8.76% | 10.09% | 28.49% | 99.32% |
| 0.3-Bottom | 10.91% | 36.44% | 16.04% | 14.55% | 39.57% | 99.28% |
| DenseNet-121 | 0.48% | 29.81% | 0.02% | 1.64% | 9.90% | 99.48% |
| 0.5-Bottom | 2.57% | 35.85% | 1.10% | 3.93% | 16.29% | 99.46% |
| 0.3-Bottom | 7.13% | 58.92% | 3.37% | 11.39% | 32.69% | 99.38% |
| SENet-18 | 0.22% | 18.77% | 2.34% | 0.10% | 13.75% | 99.41% |
| 0.5-Bottom | 3.37% | 24.09% | 6.90% | 2.21% | 17.24% | 99.35% |
| 0.3-Bottom | 11.97% | 51.63% | 14.39% | 16.45% | 40.23% | 99.31% |
| VGG-19 | 3.83% | 51.77% | 5.59% | 7.34% | 3.25% | 99.48% |
| 0.5-Bottom | 12.47% | 61.18% | 12.91% | 21.71% | 19.32% | 99.37% |
| 0.3-Bottom | 29.14% | 70.59% | 31.55% | 41.87% | 47.65% | 99.33% |

Table 9: Black-box defense performances against transfer-based attacks on the MNIST dataset. Numbers in the middle mean the success defense rates. Networks in the first row are the source models for generating adversarial examples by PGD. 0.5-Bottom and 0.3-Bottom mean applying defective convolutional layers with keep probability 0.5 and 0.3 to the bottom layers of the network whose name lies just above them. The source and target networks are initialized differently if they share the same architecture.

A.6    GRAY-BOX ATTACK

In this subsection, we show the gray-box defense performance of defective CNNs on CIFAR-10. We use gray-box attacks in the following two ways. One way is to generate adversarial examples against one trained neural network and test those images on a network with the same structure but different initializations. The other way is specific to our defective models. We generate adversarial examples on one trained defective CNN and test them on a network with the same keep probability but different sampling of defective neurons. In both of these two ways, the adversarial knows some information on the structure of the network but does not know the specific parameters of it.

| Architecture | 0.5-Bottom | 0.3-Bottom |
|---|---|---|
| 0.5-Bottom | 30.90% | 40.49% |
| 0.5-Bottom$_{DIF}$ | 32.77% | 40.39% |
| 0.3-Bottom | 59.24% | 36.84% |
| 0.3-Bottom$_{DIF}$ | 57.45% | 37.04% |

Table 10: Defense performances against two kinds of gray-box attacks for defective CNNs. Numbers mean the success defense rates. Networks in the first row are the source models for generating adversarial examples by PGD, which runs for 20 steps with step size 1 and perturbation scale $\ell_\infty = 16/255$. 0.5-Bottom and 0.3-Bottom in the left column represent the networks with the same structure as the corresponding source networks but with different initialization. 0.5-Bottom$_{DIF}$ and 0.3-Bottom$_{DIF}$ in the left column represent the networks with the same keep probabilities as the corresponding source networks but with different sampling of defective neurons.

From the results listed in Table 10, we find that defective CNNs have similar performance on adversarial examples generated by our two kinds of gray-box attacks. This phenomenon indicates that defective CNNs with the same keep probability would catch similar information which is insensitive to the selections of defective neurons. Also, comparing with the gray-box performance of standard CNNs (See Table 11), defective CNNs show stronger defense ability.

| Architecture | ResNet-18 | DenseNet-121 |
|---|---|---|
| ResNet-18 | 5.98% | 14.14% |
| DenseNet-121 | 14.53% | 2.98% |

Table 11: Defense performances against gray-box attacks for standard CNNs. Numbers mean the success defense rates. Networks in the first row are the source models for generating adversarial examples by PGD, which runs for 20 steps with step size 1 and perturbation scale $\ell_\infty = 16/255$. The diagonal shows gray-box performances in the setting that the source and target networks share the same structure but with different initializations.

## A.7 WHITE-BOX ATTACK

In this subsection, we show the white-box defense performance of defective CNNs. Table 12 shows the results of ResNet-18 on the CIFAR-10 dataset. The performance on other network architectures is similar. Note that, the proposed method would not involve any obfuscated gradients (Athalye et al., 2018). Also, We study the combination of the proposed method and adversarial training. We adversarially train a defective CNN under the same setting described in Madry et al. (2017) and reach 51.6% successful defense rate against the default PGD attack ($\ell_\infty = 8/255$ and 7 steps) used in training, which outperforms the standard CNN (50.0%).

| Architecture | FGSM$_1$ | FGSM$_2$ | FGSM$_4$ | PGD$_2$ | PGD$_4$ | PGD$_8$ | Test Accuracy |
|---|---|---|---|---|---|---|---|
| ResNet-18 | 81.24% | 65.78% | 51.24% | 23.80% | 3.16% | 0.02% | 95.33% |
| 0.5-Bottom | 85.22% | 68.65% | 52.04% | 38.16% | 6.67% | 0.12% | 93.39% |
| 0.3-Bottom | 85.70% | 69.69% | 54.51% | 49.01% | 18.93% | 2.86% | 91.83% |

Table 12: Defense performances against white-box attacks. Numbers in the middle mean the success defense rates. FGSM$_1$, FGSM$_2$, FGSM$_4$ refer to FGSM with perturbation scale 1,2,4 respectively. PGD$_2$, PGD$_4$, PGD$_8$ refer to PGD with perturbation scale 2,4,8 and step number 4,6,10 respectively. The step size of all PGD methods are set to 1.

We want to emphasize that the adversarial examples generated by defective CNNs appear to have semantic shapes and may even fool humans as well (see Figure 3 and Appendix B). This indicates that small perturbations can actually change the semantic meaning of images for humans. Those samples should probably not be categorized into adversarial examples and used to evaluate white-box robustness. This is also aligned with Ilyas et al. (2019).

## A.8 RANDOMLY SHUFFLED IMAGES AND STYLIZED-IMAGENET

In this subsection, we show more results on randomly shuffled images and Stylized-ImageNet (Geirhos et al., 2018). As shown in Section 3.2, shape information in randomly shuffled images is destroyed while textural information preserving, and Stylized-ImageNet has the opposite situation. If a CNN make predictions relying less on textural information but more on shape information, it should have worse performance on randomly shuffled images but better performance on Stylized-ImageNet.

We construct defective CNNs by applying defective convolutional layers to the bottom layers of standard ResNet-18, ResNet-50, DenseNet-121, SENet-18, and VGG-19. We train all defective CNNs and their plain counterparts on the ImageNet dataset. For each pair of CNNs, we sample images from the validation set, which are predicted correctly by both two kinds of CNNs. We make manipulations to the sampled images by setting $k \in \{2, 4, 8\}$ and pick corresponding images from Stylized-ImageNet. We check the accuracy of all models on the these images. The results are shown in Table 13. We can see the defective CNNs perform consistently worse than the standard CNNs on

the randomly shuffled images, and perform consistently better than the standard CNNs on Stylized-ImageNet. This justifies our argument that defective CNNs make predictions relying less on textural information but more on shape information.

| Model | $2 \times 2$ | $4 \times 4$ | $8 \times 8$ | IN $\rightarrow$ SIN |
|---|---|---|---|---|
| ResNet-18 | 99.53% | 84.36% | 20.08% | 15.33% |
| 0.1-Bottom | 96.32% | 56.91% | 9.04% | 20.20% |
| ResNet-50 | 99.80% | 87.34% | 18.00% | 15.12% |
| 0.1-Bottom | 98.30% | 65.87% | 9.23% | 21.16% |
| DenseNet-121 | 99.55% | 85.87% | 18.78% | 15.53% |
| 0.1-Bottom | 92.23% | 47.82% | 7.52% | 19.09% |
| SENet-18 | 98.88% | 75.57% | 14.61% | 15.83% |
| 0.1-Bottom | 92.39% | 46.94% | 7.07% | 18.79% |
| VGG-19 | 99.10% | 81.58% | 15.45% | 6.17% |
| 0.1-Bottom | 97.98% | 71.85% | 11.00% | 13.98% |

Table 13: The left three columns are the accuracy of classifying randomly shuffled test images. The rightmost column is the accuracy of training on ImageNet and testing on Stylized-ImageNet. 0.1-Bottom mean applying defective convolutional layers with keep probability 0.1 to the bottom layers of the network whose name lies just above them.

## A.9 DIFFERENT KEEP PROBABILITIES

In this subsection, we show the trade-off between robustness and generalization performance in defective CNNs with different keep probabilities. We use DenseNet-121 (Huang et al., 2017) as the source model to generate adversarial examples from CIFAR-10 with PGD (Kurakin et al., 2016), which runs for 20 steps with step size 1 and perturbation scale 16. The defective convolutional layers are applied to the bottom layers of ResNet-18 (He et al., 2016). Figure 5 shows that the defense accuracy monotonically increases as the test accuracy decreases along with the keep probability. We can see the trade-off between robustness and generalization.

Therefore, a practical way to use Defective CNNs in the real world is to ensemble defective CNNs with different keep probabilities. Also, in our experiments, we found that ensemble different defective CNNs with the same p can bring improvements on both accuracy and robustness while ensemble standard CNNs can not.

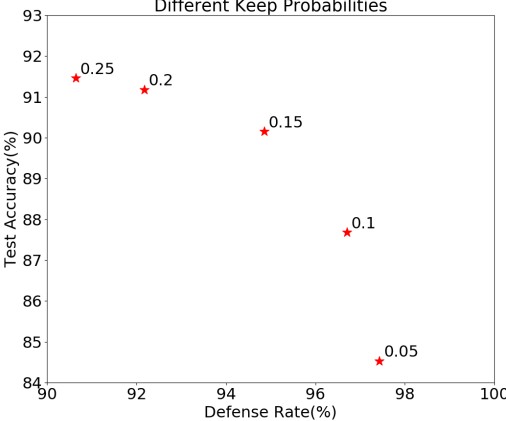

Figure 5: Relationship between success defense rates against adversarial examples generated by PGD and test accuracy with respect to different keep probabilities. Each red star represents a specific keep probability with its value written near the star.

## A.10 EXPERIMENTAL DETAILS FOR SECTION 4.3

In this subsection, we will show more experimental results on defective CNNs using different adversarial examples, different attack methods and different mask settings on ResNet-18. The networks used to generate adversarial examples including ResNet-18, ResNet-50, DenseNet-121, SENet18, and VGG-19. More specifically, we choose 5000 samples to generate adversarial examples via FGSM and PGD, and 1000 samples for CW attack. All samples are drawn from the validation set of CIFAR-10 dataset and can be correctly classified correctly by the model used to generate adversarial examples.

For FGSM, we try step size $\epsilon \in \{8, 16, 32\}$, namely **FGSM$_8$**, **FGSM$_{16}$**, **FGSM$_{32}$**, to generate adversarial examples. For PGD, we have tried more extensive settings. Let $\{\epsilon, T, \alpha\}$ be the PGD setting with step size $\epsilon$, the number of steps $T$ and the perturbation scale $\alpha$, then we have tried PGD settings $(1, 8, 4), (2, 4, 4), (4, 2, 4), (1, 12, 8), (2, 6, 8), (4, 3, 8), (1, 20, 16), (2, 10, 16), (4, 5, 16),$ $(1, 40, 32), (2, 20, 32), (4, 10, 32)$ to generate PGD adversarial examples. From the experimental results, we observe the following phenomena. First, we find that the larger the perturbation scale is, the stronger the adversarial examples are. Second, for a fixed perturbation scale, the smaller the step size is, the more successful the attack is, as it searches the adversarial examples in a more careful way around the original image. Based on these observation, we only show strong PGD attack results in the Appendix, namely the settings $(1, 20, 16)$ (**PGD$_{16}$**), $(2, 10, 16)$ (**PGD$_{2,16}$**) and $(1, 40, 32)$ (**PGD$_{32}$**). Nonetheless, our models also perform much better on weak PGD attacks. For the CW attack, we have also tried different confidence parameters $\kappa$. However, we find that for large $\kappa$, the algorithm is hard to find adversarial examples for some neural networks such as VGG because of its logit scale. For smaller $\kappa$, the adversarial examples have weak transferability, which means they can be easily defended even by standard CNNs. Therefore, in order to balance these two factors, we choose $\kappa = 40$ (**CW$_{40}$**) for DenseNet-121, ResNet-50, SENet-18 and $\kappa = 20$ (**CW$_{20}$**) for ResNet-18 as a good choice to compare our models with standard ones. The step number for choosing the parameter $c$ is set to 30.

Note that the noises of FGSM and PGD are considered in the sense of $\ell_\infty$ norm and the noise of CW is considered in the sense of $\ell_2$ norm. All adversarial examples used to evaluate can fool the original network. Table 14,15,16,17 and 18 list our experimental results. DC means we replace defective neurons with defective channels in the corresponding blocks to achieve the same keep probability. SM means we use the same defective mask on all the channels in a layer. $\times n$ means we multiply the number of the channels in the defective blocks by $n$ times. EN means we ensemble five models with different defective masks of the same keep probability.

## B ADVERSARIAL EXAMPLES GENERATED BY DEFECTIVE CNNS

In this subsection, we show more adversarial examples generated by defective CNNs. Figure 6 shows some adversarial examples generated on the CIFAR-10 dataset along with the corresponding original images. These examples are generated from CIFAR-10 against a defective ResNet-18 of keep probability 0.2 on the $0^{th}, 1^{st}, 2^{nd}$ blocks, a defective ResNet-18 of keep probability 0.1 on the $1^{st}, 2^{nd}$ blocks, and a standard ResNet-18. We use attack method MIFGSM (Dong et al., 2017) with perturbation scale $\alpha = 16$ and $\alpha = 32$. We also show some adversarial examples generated from Tiny-ImageNet[3] along with the corresponding original images in Figure 7. These examples are generated from Tiny-ImageNet against a defective ResNet-18 of keep probability of the keep probability 0.1 on the $1^{st}, 2^{nd}$ blocks and a standard ResNet-18. The attack methods are MIFGSM with scale 64 and 32, step size 1 and step number 40 and 80 respectively.

The adversarial examples generated by defective CNNs exhibit more semantic shapes of their fooled classes, such as the mouth of the frog in Figure 6. This also corroborates the point made in Tsipras et al. (2018) that more robust models will be more aligned with human perception.

To further verify the adversarial examples generated by defective CNNs align better with human perception than standard CNNs, we conduct a user study. We show users a pair of adversarial examples generated by defective CNNs and standard CNNs, respectively. The corresponding labels are attached. The user will be asked which one of the pair is better aligned with the predicted label.

---

[3]https://tiny-imagenet.herokuapp.com/

More specifically, we generate two sets of adversarial examples on CIFAR-10 and Tiny-ImageNet by defective CNNs and standard CNNs, respectively. For each user, we randomly sample 50 pairs from the two sets and ask him/her to select. A total of 13 people are involved in our study. The results show that all users select more images generated by defective CNNs than the ones generated by standard CNNs. On average, the number of defective CNNs ones is 14 more than the number of standard CNNs ones. This supports our arguments.

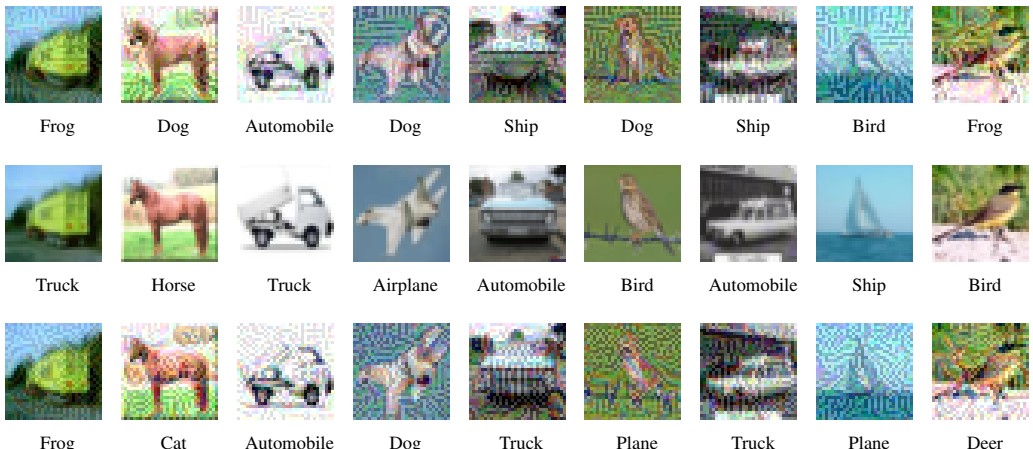

Figure 6: CIFAR-10 dataset. **First row**: the adversarial examples generated by defective CNNs and the predicted labels. **Second row**: original images. **Third row**: the adversarial examples generated by the standard CNN and the predicted labels.

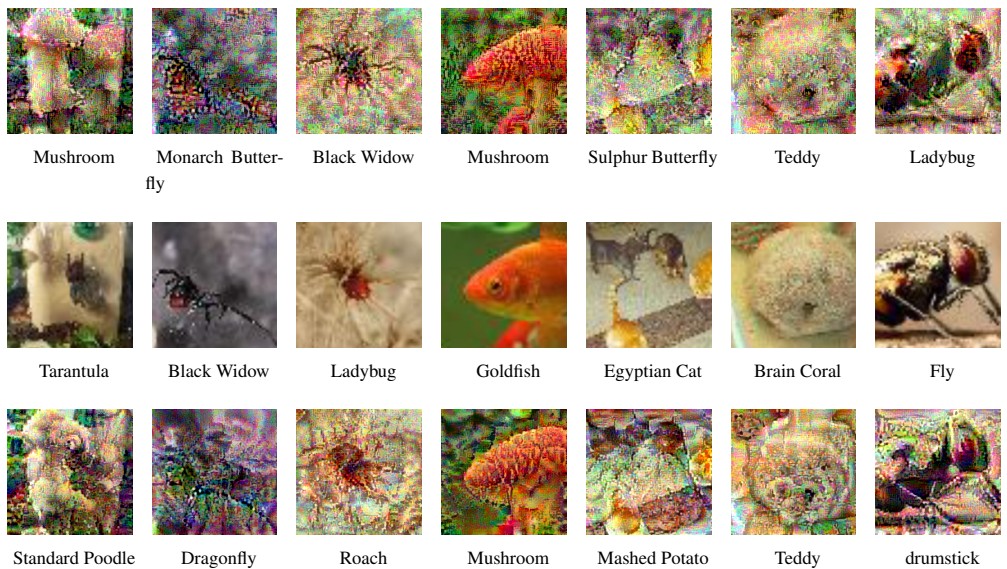

Figure 7: Tiny-ImageNet dataset. **First row**: the adversarial examples generated by defective CNNs and the predicted labels. **Second row**: original images. **Third row**: the adversarial examples generated by the standard CNN and the predicted labels.

## C    ARCHITECTURE ILLUSTRATIONS

In this subsection, we briefly introduce the network architectures used in our experiments. Generally, we apply defective convolutional layers to the bottom layers of the networks and we have tried six different architectures, namely **ResNet-18**, **ResNet-50**, **DenseNet-121**, **SENet-18**, **VGG-19** and **WideResNet-32**. We next illustrate these architectures and show how we apply defective convolutional layers to them. In our experiments, applying defective convolutional layers to a block means randomly selecting defective neurons in every layer of the block.

### C.1    RESNET-18

ResNet-18 (He et al., 2016) contains 5 blocks: the $0^{\text{th}}$ block is one single $3 \times 3$ convolutional layer, and each of the rest contains four $3 \times 3$ convolutional layers. Figure 8 shows the whole structure of ResNet-18. In our experiments, we apply defective convolutional layers to the $0^{\text{th}}, 1^{\text{st}}, 2^{\text{nd}}$ blocks which are the bottom layers.

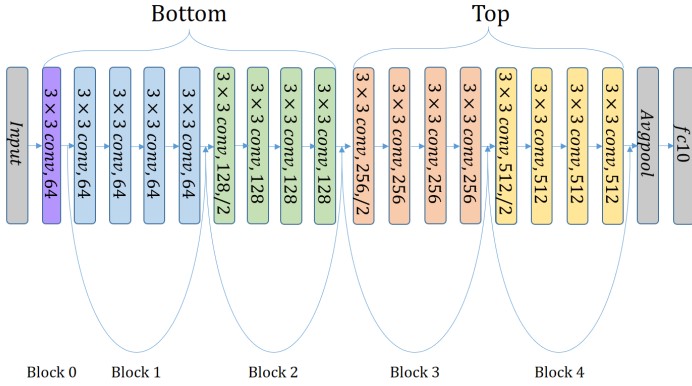

Figure 8: The architecture of ResNet-18

### C.2    RESNET-50

Similar to ResNet-18, ResNet-50 (He et al., 2016) contains 5 blocks and each block contains several $1 \times 1$ and $3 \times 3$ convolutional layers (i.e. Bottlenecks). In our experiment, we apply defective convolutional layers to the $3 \times 3$ convolutional layers in the first three "bottom" blocks. The defective layers in the $1^{\text{st}}$ block are marked by the red arrows in Figure 9.

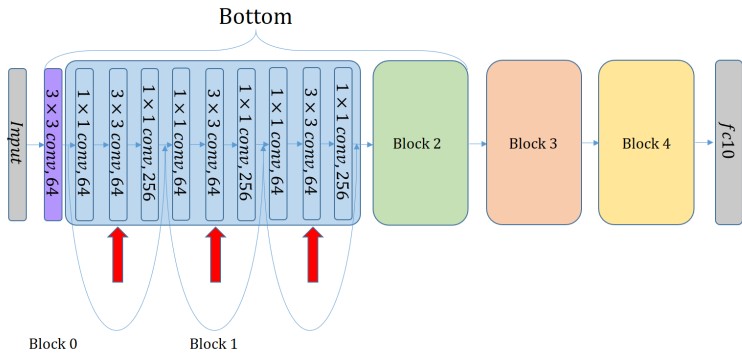

Figure 9: The architecture of ResNet-50

### C.3 DENSENET-121

DenseNet-121 (Huang et al., 2017) is another popular network architecture in deep learning researches. Figure 10 shows the whole structure of DenseNet-121. It contains 5 Dense-Blocks, each of which contains several $1 \times 1$ and $3 \times 3$ convolutional layers. Similar to what we do for ResNet-50, we apply defective convolutional layers to the $3 \times 3$ convolutional layers in the first three "bottom" blocks. The growth rate is set to 32 in our experiments. vspace-0.1in

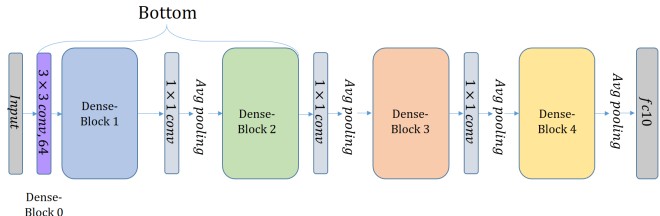

Figure 10: The architecture of DenseNet-121

### C.4 SENET-18

SENet (Hu et al., 2017a), a network architecture which won the first place in ImageNet contest 2017, is shown in Figure 11. Note that here we use the pre-activation shortcut version of SENet-18 and we apply defective convolutional layers to the convolutional layers in the first 3 SE-blocks.

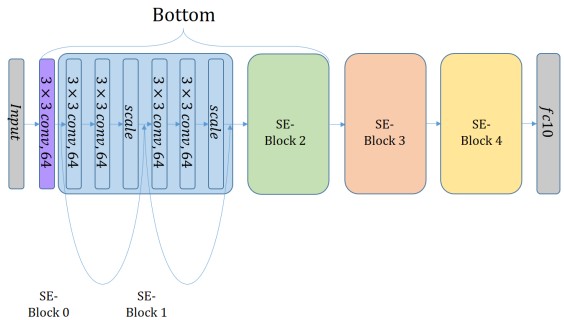

Figure 11: The architecture of SENet-18

### C.5 VGG-19

VGG-19 (Simonyan & Zisserman, 2014) is a typical neural network architecture with sixteen $3 \times 3$ convolutional layers and three fully-connected layers. We slightly modified the architecture by replacing the final 3 fully connected layers with 1 fully connected layer as is suggested by recent architectures. Figure 12 shows the whole structure of VGG-19. We apply defective convolutional layers on the first four $3 \times 3$ convolutional layers.

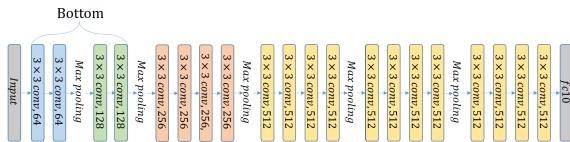

Figure 12: The architecture of VGG-19

## C.6 WideResNet-32

Based on residual networks, Zagoruyko & Komodakis (2016) proposed a wide version of residual networks which have much more channels. In our experiments, we adopt the network with a width factor of 4 and apply defective layers on the 0th and 1st blocks. Figure 13 shows the whole structure of WideResNet-32.

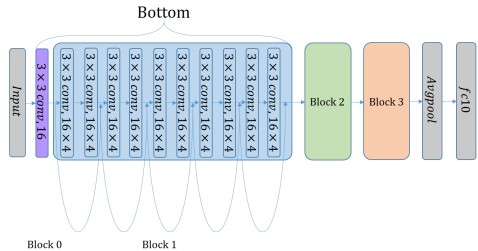

Figure 13: The architecture of WideResNet-32

## D    Training Details on CIFAR-10 and MNIST

To guarantee our experiments are reproducible, here we present more details on the training process in our experiments. When training models on CIFAR-10, we first subtract per-pixel mean. Then we apply a zero-padding of width 4, a random horizontal flip and a random crop of size $32 \times 32$ on train data. No other data augmentation method is used. We apply SGD with momentum parameter 0.9, weight decay parameter $5 \times 10^{-4}$ and mini-batch size 128 to train on the data for 350 epochs. The learning rate starts from 0.1 and is divided by 10 when the number of epochs reaches 150 and 250. When training models on MNIST, we first subtract per-pixel mean. Then we apply random horizontal flip on train data. We apply SGD with momentum parameter 0.9, weight decay parameter $5 \times 10^{-4}$ and mini-batch size 128 to train on the data for 50 epochs. The learning rate starts from 0.1 and is divided by 10 when the number of epochs reaches 20 and 40. Figure 14 shows the train and test curves of standard and defective ResNet-18 on CIFAR-10 and MNIST. Different network structures share similar tendency regarding the train and test curves.

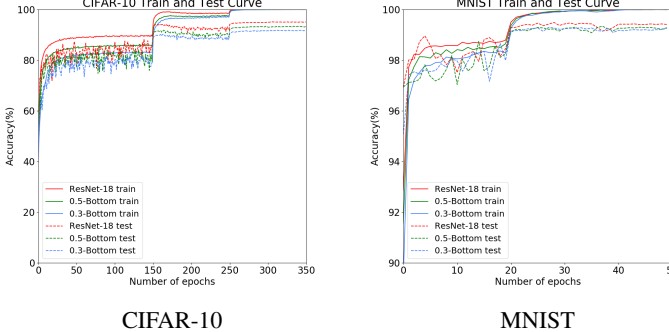

Figure 14: Train and test curve of standard and defective ResNet-18 on CIFAR-10 and MNIST

## E    Attack approaches

In this subsection, we describe the attack approaches used in our experiments. We first give an overview of how to attack a neural network in mathematical notations. Let $x$ be the input to the neural network and $f_{\boldsymbol{\theta}}$ be the function which represents the neural network with parameter $\boldsymbol{\theta}$. The output label of the network to the input can be computed as $c = \arg\max_i f_{\boldsymbol{\theta}}(\boldsymbol{x})_i$. In order to perform an adversarial attack, we add a small perturbation $\delta_x$ to the original image and get an

adversarial image $\boldsymbol{x}_{\text{adv}} = \boldsymbol{x} + \delta_x$. The new input $\boldsymbol{x}_{\text{adv}}$ should look visually similar to the original $\boldsymbol{x}$. Here we use the commonly used $\ell_\infty$-norm metric to measure similarity, i.e., we require that $||\delta_x|| \leq \epsilon$. The attack is considered successful if the predicted label of the perturbed image $c_{\text{adv}} = \arg\max_i f_\theta(\boldsymbol{x}_{\text{adv}})_i$ is different from $c$.

Generally speaking, there are two types of attack methods: *Targeted Attack*, which aims to change the output label of an image to a specific (and different) one, and *Untargeted Attack*, which only aims to change the output label and does not restrict which specific label the modified example should let the network output.

In this paper, we mainly use the following four gradient-based attack approaches. $J$ denotes the loss function of the neural network and $y$ denotes the ground truth label of $\boldsymbol{x}$.

- **Fast Gradient Sign Method (FGSM).** FGSM (Goodfellow et al., 2015) is a one-step untargeted method which generates the adversarial example $\boldsymbol{x}_{\text{adv}}$ by adding the sign of the gradients multiplied by a step size $\epsilon$ to the original benign image $\boldsymbol{x}$. Note that FGSM controls the $\ell_\infty$-norm between the adversarial example and the original one by the parameter $\epsilon$.
$$\boldsymbol{x}_{\text{adv}} = \boldsymbol{x} + \epsilon \cdot \text{sign}(\nabla_{\boldsymbol{x}} J(\boldsymbol{x}, y)).$$

- **Basic iterative method (PGD).** PGD (Kurakin et al., 2016) is a multiple-step attack method which applies FGSM multiple times. To make the adversarial example still stay "close" to the original image, the image is projected to the $\ell_\infty$-ball centered at the original image after every step. The radius of the $\ell_\infty$-ball is called perturbation scale and is denoted by $\alpha$.
$$\boldsymbol{x}_{\text{adv}}^0 = \boldsymbol{x}, \quad \boldsymbol{x}_{\text{adv}}^{k+1} = \text{Clip}_{\boldsymbol{x},\alpha} \left[ \boldsymbol{x}_{\text{adv}}^k + \epsilon \cdot \text{sign}(\nabla_{\boldsymbol{x}} J(\boldsymbol{x}_{\text{adv}}^k, y)) \right].$$

- **Momentum Iterative Fast Gradient Sign Method (MIFGSM).**
MIFGSM (Dong et al., 2017) is a recently proposed multiple-step attack method. It is similar to PGD, but it computes the optimize direction by a momentum instead of the gradients. The radius of the $\ell_\infty$-ball is also called perturbation scale and is denoted by $\alpha$.

$$g_{k+1} = \mu \cdot g_k + \frac{\nabla_{\boldsymbol{x}} J(\boldsymbol{x}_{\text{adv}}^k, y)}{\|\nabla_{\boldsymbol{x}} J(\boldsymbol{x}_{\text{adv}}^k, y)\|_1}$$

$$\boldsymbol{x}_{\text{adv}}^0 = \boldsymbol{x}, g_0 = 0 \quad \boldsymbol{x}_{\text{adv}}^{k+1} = \text{Clip}_{\boldsymbol{x},\alpha} \left[ \boldsymbol{x}_{\text{adv}}^k + \epsilon \cdot \text{sign}(g_{k+1}) \right].$$

- **CW Attack.** Carlini & Wagner (2016) shows that constructing an adversarial example can be formulated as solving the following optimization problem:
$$\boldsymbol{x}_{\text{adv}} = \arg\min_{\boldsymbol{x}'} c \cdot g(\boldsymbol{x}') + ||\boldsymbol{x}' - \boldsymbol{x}||_2^2,$$

where $c \cdot g(\boldsymbol{x}')$ is the loss function that evaluates the quality of $\boldsymbol{x}'$ as an adversarial example and the term $||\boldsymbol{x}' - \boldsymbol{x}||_2^2$ controls the scale of the perturbation. More specifically, in the untargeted attack setting, the loss function $g(\boldsymbol{x})$ can be defined as below, where the parameter $\kappa$ is called confidence.

$$g(\boldsymbol{x}) = \max\{\max_{i \neq y} (f(\boldsymbol{x})_i) - f(\boldsymbol{x})_y, -\kappa\},$$

| Architecture | $FGSM_8$ | $FGSM_{16}$ | $FGSM_{32}$ | $PGD_{16}$ | $PGD_{2,16}$ | $PGD_{32}$ | $CW_{40}$ | Acc |
|---|---|---|---|---|---|---|---|---|
| ResNet-18 | 29.78% | 14.91% | 11.53% | 14.14% | 10.02% | 7.16% | 8.23% | 95.33% |
| 0.7-Bottom | 55.40% | 23.29% | 7.73% | 51.29% | 42.44% | 37.00% | 36.95% | 94.03% |
| 0.5-Bottom | 66.87% | 30.86% | 6.65% | 70.38% | 62.11% | 56.36% | 54.02% | 93.39% |
| 0.3-Bottom | 79.50% | 48.57% | 10.51% | 86.99% | 81.78% | 78.41% | 73.70% | 91.83% |
| 0.25-Bottom | 83.12% | 59.22% | 17.16% | 90.64% | 86.22% | 83.86% | 77.82% | 91.46% |
| 0.2-Bottom | 85.49% | 63.01% | 15.57% | 92.17% | 88.72% | 86.50% | 81.75% | 91.18% |
| 0.15-Bottom | 88.18% | 65.27% | 18.33% | 94.85% | 92.24% | 90.64% | 85.46% | 90.15% |
| 0.1-Bottom | 94.08% | 79.93% | 43.70% | 96.70% | 95.69% | 94.68% | 89.67% | 87.68% |
| 0.05-Bottom | 96.16% | 87.36% | 59.05% | 97.43% | 97.13% | 96.24% | 90.24% | 84.53% |
| 0.7-Top | 28.51% | 14.62% | 8.78% | 10.55% | 7.22% | 4.91% | 7.88% | 95.16% |
| 0.5-Top | 25.01% | 10.76% | 10.24% | 11.06% | 7.99% | 5.10% | 7.19% | 94.94% |
| 0.3-Top | 23.94% | 11.23% | 10.48% | 11.77% | 8.83% | 5.80% | 10.10% | 94.61% |
| 0.5-Bottom, 0.5-Top | 55.88% | 20.77% | 9.96% | 60.75% | 51.47% | 45.29% | 47.32% | 92.48% |
| 0.7-Bottom, 0.7-Top | 51.03% | 24.15% | 10.82% | 45.12% | 35.70% | 30.24% | 29.65% | 94.16% |
| 0.7-Bottom, 0.3-Top | 36.16% | 11.26% | 9.16% | 33.67% | 24.62% | 20.28% | 23.31% | 93.44% |
| 0.3-Bottom, 0.3-Top | 64.85% | 27.43% | 10.09% | 75.49% | 67.09% | 62.78% | 62.47% | 89.78% |
| 0.3-Bottom, 0.7-Top | 74.73% | 40.58% | 9.12% | 82.77% | 75.98% | 72.15% | 68.58% | 91.23% |
| $0.5\text{-Bottom}_{DC}$ | 36.39% | 12.15% | 8.24% | 19.93% | 14.99% | 11.20% | 12.72% | 95.12% |
| $0.3\text{-Bottom}_{DC}$ | 43.81% | 17.74% | 8.32% | 27.47% | 21.73% | 16.59% | 19.34% | 94.23% |
| $0.1\text{-Bottom}_{DC}$ | 49.53% | 19.00% | 7.23% | 53.87% | 44.78% | 41.40% | 44.80% | 93.27% |
| $0.5\text{-Bottom}_{SM}$ | 77.30% | 48.86% | 12.50% | 85.00% | 80.01% | 75.60% | 72.07% | 92.57% |
| $0.3\text{-Bottom}_{SM}$ | 82.59% | 48.03% | 12.30% | 91.01% | 87.35% | 84.71% | 79.55% | 89.81% |
| $0.1\text{-Bottom}_{SM}$ | 67.06% | 39.40% | 16.25% | 80.36% | 74.97% | 72.43% | 65.38% | 74.28% |
| $0.5\text{-Bottom}_{\times 2}$ | 51.25% | 20.78% | 10.29% | 50.16% | 40.47% | 34.24% | 34.00% | 94.12% |
| $0.3\text{-Bottom}_{\times 2}$ | 68.82% | 30.94% | 7.22% | 76.62% | 67.90% | 62.87% | 60.17% | 93.01% |
| $0.1\text{-Bottom}_{\times 2}$ | 88.00% | 68.83% | 28.55% | 93.35% | 90.82% | 88.25% | 82.74% | 90.49% |
| $ResNet\text{-}18_{EN}$ | 34.98% | 16.51% | 10.32% | 12.60% | 9.22% | 5.48% | 8.46% | 96.03% |
| $0.5\text{-Bottom}_{\times 2,EN}$ | 58.49% | 20.75% | 8.48% | 57.47% | 47.05% | 39.21% | 41.36% | 95.10% |
| $0.5\text{-Bottom}_{EN}$ | 69.35% | 31.38% | 7.73% | 75.40% | 66.37% | 60.59% | 58.07% | 94.56% |
| $0.3\text{-Bottom}_{EN}$ | 81.98% | 51.81% | 8.57% | 90.00% | 85.25% | 82.15% | 77.74% | 93.31% |
| $0.1\text{-Bottom}_{EN}$ | 95.37% | 81.95% | 43.42% | 97.91% | 97.10% | 95.90% | 91.36% | 89.45% |

Table 14: Extended experimental results of Section 4.3. Adversarial examples generated against *DenseNet-121*. Numbers in the middle mean the success defense rates. The model trained on CIFAR-10 achieves 95.62% accuracy on test set. $p$-Bottom, $p$-Top, $p$-Bottom$_{DC}$, $p$-Bottom$_{SM}$, $p$-Bottom$_{\times n}$ and $p$-Bottom$_{EN}$ mean applying defective layers with keep probability $p$ to bottom layers, applying defective layers with keep probability $p$ to top layers, making whole channels defective with keep probability $p$, using the same defective mask in every channel with keep probability $p$, increasing channel number to $n$ times at bottom layers and ensemble five models with different defective masks of the same keep probability $p$ respectively.

| Architecture | $FGSM_8$ | $FGSM_{16}$ | $FGSM_{32}$ | $PGD_{16}$ | $PGD_{2,16}$ | $PGD_{32}$ | $CW_{20}$ | Acc |
|---|---|---|---|---|---|---|---|---|
| ResNet-18 | 26.99% | 13.91% | 3.57% | 5.98% | 3.70% | 3.02% | 2.19% | 95.33% |
| 0.7-Bottom | 48.76% | 21.32% | 9.54% | 34.43% | 25.14% | 24.16% | 38.87% | 94.03% |
| 0.5-Bottom | 59.66% | 30.48% | 11.60% | 53.89% | 45.47% | 41.27% | 60.65% | 93.39% |
| 0.3-Bottom | 74.00% | 47.11% | 15.65% | 78.23% | 73.30% | 65.83% | 79.04% | 91.83% |
| 0.25-Bottom | 78.37% | 56.05% | 21.44% | 83.96% | 80.09% | 73.45% | 81.59% | 91.46% |
| 0.2-Bottom | 81.67% | 59.14% | 19.60% | 88.18% | 85.07% | 78.72% | 82.78% | 91.18% |
| 0.15-Bottom | 86.31% | 63.16% | 22.23% | 92.26% | 89.99% | 85.14% | 86.06% | 90.15% |
| 0.1-Bottom | 92.89% | 77.90% | 45.63% | 96.27% | 95.30% | 92.80% | 90.29% | 87.68% |
| 0.05-Bottom | 95.07% | 85.40% | 59.91% | 97.51% | 96.69% | 95.30% | 90.97% | 84.53% |
| 0.7-Top | 25.96% | 15.46% | 7.18% | 5.36% | 2.83% | 2.89% | 2.66% | 95.16% |
| 0.5-Top | 25.21% | 9.21% | 1.44% | 5.98% | 4.30% | 3.25% | 3.15% | 94.94% |
| 0.3-Top | 24.36% | 9.49% | 2.60% | 8.54% | 5.30% | 5.02% | 6.62% | 94.61% |
| 0.5-Bottom, 0.5-Top | 51.89% | 20.41% | 10.75% | 45.99% | 37.78% | 34.09% | 52.11% | 92.48% |
| 0.7-Bottom, 0.7-Top | 43.32% | 20.55% | 4.14% | 29.28% | 19.64% | 20.14% | 32.92% | 94.16% |
| 0.7-Bottom, 0.3-Top | 34.09% | 11.05% | 1.58% | 23.06% | 15.26% | 14.93% | 24.11% | 93.44% |
| 0.3-Bottom, 0.3-Top | 61.22% | 28.11% | 13.78% | 67.30% | 59.43% | 52.95% | 69.15% | 89.78% |
| 0.3-Bottom, 0.7-Top | 70.43% | 39.15% | 13.94% | 74.85% | 68.23% | 62.52% | 74.57% | 91.23% |
| $0.5\text{-Bottom}_{DC}$ | 32.86% | 13.89% | 3.71% | 9.41% | 5.60% | 5.34% | 6.10% | 95.12% |
| $0.3\text{-Bottom}_{DC}$ | 37.96% | 16.23% | 5.05% | 16.63% | 11.49% | 10.54% | 15.44% | 94.23% |
| $0.1\text{-Bottom}_{DC}$ | 48.54% | 19.10% | 11.37% | 41.14% | 31.82% | 30.56% | 50.62% | 93.27% |
| $0.5\text{-Bottom}_{SM}$ | 73.96% | 47.63% | 16.60% | 75.60% | 68.88% | 62.10% | 73.68% | 92.57% |
| $0.3\text{-Bottom}_{SM}$ | 80.80% | 48.37% | 15.26% | 87.88% | 84.37% | 77.69% | 82.34% | 89.81% |
| $0.1\text{-Bottom}_{SM}$ | 69.15% | 43.55% | 20.26% | 79.96% | 75.52% | 71.95% | 71.62% | 74.28% |
| $0.5\text{-Bottom}_{\times 2}$ | 46.50% | 21.37% | 6.06% | 32.98% | 22.90% | 22.66% | 39.12% | 94.12% |
| $0.3\text{-Bottom}_{\times 2}$ | 63.37% | 29.90% | 12.07% | 61.81% | 53.25% | 48.36% | 67.02% | 93.01% |
| $0.1\text{-Bottom}_{\times 2}$ | 84.28% | 64.47% | 31.90% | 90.76% | 87.81% | 82.61% | 85.08% | 90.49% |
| $\text{ResNet-18}_{EN}$ | 29.36% | 13.89% | 3.81% | 4.72% | 2.88% | 2.08% | 2.09% | 96.03% |
| $0.5\text{-Bottom}_{\times 2,EN}$ | 51.63% | 20.74% | 7.58% | 37.99% | 26.65% | 26.61% | 42.59% | 95.10% |
| $0.5\text{-Bottom}_{EN}$ | 63.38% | 30.25% | 11.05% | 56.29% | 46.76% | 42.75% | 63.90% | 94.56% |
| $0.3\text{-Bottom}_{EN}$ | 77.25% | 50.07% | 13.80% | 80.40% | 75.52% | 68.16% | 80.86% | 93.31% |
| $0.1\text{-Bottom}_{EN}$ | 94.31% | 79.47% | 44.67% | 97.20% | 95.90% | 94.03% | 90.52% | 89.45% |

Table 15: Extended experimental results of Section 4.3. Numbers in the middle mean the success defense rates. Adversarial examples are generated against *ResNet-18*. The model trained on CIFAR-10 achieves 95.27% accuracy on test set. $p$-Bottom, $p$-Top, $p\text{-Bottom}_{DC}$, $p\text{-Bottom}_{SM}$, $p\text{-Bottom}_{\times n}$ and $p\text{-Bottom}_{EN}$ mean applying defective layers with keep probability $p$ to bottom layers, applying defective layers with keep probability $p$ to top layers, making whole channels defective with keep probability $p$, using the same defective mask in every channel with keep probability $p$, increasing channel number to $n$ times at bottom layers and ensemble five models with different defective masks of the same keep probability $p$ respectively.

| Architecture | $FGSM_8$ | $FGSM_{16}$ | $FGSM_{32}$ | $PGD_{16}$ | $PGD_{2,16}$ | $PGD_{32}$ | $CW_{40}$ | Acc |
|---|---|---|---|---|---|---|---|---|
| ResNet-18 | 29.33% | 15.14% | 3.88% | 0.94% | 1.36% | 0.08% | 0.00% | 95.33% |
| 0.7-Bottom | 45.32% | 18.89% | 9.16% | 12.78% | 13.14% | 3.11% | 1.98% | 94.03% |
| 0.5-Bottom | 56.26% | 27.32% | 10.72% | 33.05% | 31.71% | 15.13% | 8.92% | 93.39% |
| 0.3-Bottom | 70.57% | 42.40% | 14.98% | 67.64% | 65.36% | 48.07% | 33.08% | 91.83% |
| 0.25-Bottom | 77.18% | 53.01% | 19.68% | 77.14% | 74.46% | 59.23% | 39.10% | 91.46% |
| 0.2-Bottom | 80.33% | 56.21% | 18.03% | 83.36% | 81.58% | 69.09% | 47.52% | 91.18% |
| 0.15-Bottom | 84.81% | 61.02% | 21.50% | 89.61% | 87.65% | 78.29% | 53.71% | 90.15% |
| 0.1-Bottom | 92.17% | 77.68% | 45.93% | 94.82% | 94.24% | 90.22% | 66.70% | 87.68% |
| 0.05-Bottom | 94.43% | 85.54% | 60.71% | 96.41% | 96.27% | 93.65% | 71.82% | 84.53% |
| 0.7-Top | 27.78% | 15.03% | 8.07% | 0.60% | 0.82% | 0.00% | 0.00% | 95.16% |
| 0.5-Top | 27.24% | 10.29% | 2.47% | 0.62% | 0.92% | 0.04% | 0.00% | 94.94% |
| 0.3-Top | 24.81% | 9.99% | 2.50% | 0.73% | 1.11% | 0.02% | 0.00% | 94.61% |
| 0.5-Bottom, 0.5-Top | 47.22% | 17.48% | 10.16% | 23.66% | 23.02% | 9.32% | 6.51% | 92.48% |
| 0.7-Bottom, 0.7-Top | 42.18% | 18.20% | 5.26% | 9.88% | 10.52% | 2.41% | 1.64% | 94.16% |
| 0.7-Bottom, 0.3-Top | 33.11% | 11.08% | 2.27% | 6.09% | 6.41% | 0.86% | 0.55% | 93.44% |
| 0.3-Bottom, 0.3-Top | 56.39% | 24.14% | 12.18% | 51.43% | 48.19% | 31.23% | 22.25% | 89.78% |
| 0.3-Bottom, 0.7-Top | 66.33% | 36.31% | 13.09% | 62.31% | 59.74% | 41.88% | 30.68% | 91.23% |
| $0.5\text{-Bottom}_{DC}$ | 31.56% | 13.64% | 4.87% | 1.61% | 1.81% | 0.12% | 0.11% | 95.12% |
| $0.3\text{-Bottom}_{DC}$ | 37.52% | 15.72% | 5.38% | 3.92% | 4.44% | 0.56% | 0.44% | 94.23% |
| $0.1\text{-Bottom}_{DC}$ | 44.00% | 16.90% | 10.30% | 20.34% | 20.32% | 7.95% | 4.95% | 93.27% |
| $0.5\text{-Bottom}_{SM}$ | 69.40% | 41.82% | 14.27% | 62.62% | 60.04% | 40.30% | 26.65% | 92.57% |
| $0.3\text{-Bottom}_{SM}$ | 77.25% | 44.94% | 13.80% | 81.91% | 79.90% | 67.76% | 46.44% | 89.81% |
| $0.1\text{-Bottom}_{SM}$ | 64.32% | 39.76% | 19.21% | 74.47% | 71.88% | 63.05% | 45.18% | 74.28% |
| $0.5\text{-Bottom}_{\times 2}$ | 41.51% | 18.47% | 6.02% | 10.80% | 11.40% | 2.21% | 1.32% | 94.12% |
| $0.3\text{-Bottom}_{\times 2}$ | 58.59% | 25.92% | 11.20% | 42.49% | 40.44% | 21.05% | 13.77% | 93.01% |
| $0.1\text{-Bottom}_{\times 2}$ | 83.05% | 63.73% | 29.22% | 86.47% | 84.39% | 75.07% | 50.11% | 90.49% |
| $\text{ResNet-18}_{EN}$ | 32.80% | 15.67% | 4.65% | 0.70% | 1.00% | 0.02% | 0.00% | 96.03% |
| $0.5\text{-Bottom}_{\times 2,EN}$ | 47.40% | 17.32% | 7.23% | 12.64% | 12.84% | 2.54% | 2.19% | 95.10% |
| $0.5\text{-Bottom}_{EN}$ | 59.64% | 26.21% | 10.17% | 33.55% | 32.11% | 13.93% | 8.12% | 94.56% |
| $0.3\text{-Bottom}_{EN}$ | 73.45% | 45.60% | 12.99% | 69.95% | 67.14% | 48.83% | 32.60% | 93.31% |
| $0.1\text{-Bottom}_{EN}$ | 93.87% | 79.15% | 46.44% | 96.12% | 95.82% | 91.98% | 67.71% | 89.45% |

Table 16: Extended experimental results of Section 4.3. Adversarial examples are generated against *ResNet-50*. Numbers in the middle mean the success defense rates. The model trained on CIFAR-10 achieves 95.69% accuracy on test set. $p$-Bottom, $p$-Top, $p\text{-Bottom}_{DC}$, $p\text{-Bottom}_{SM}$, $p\text{-Bottom}_{\times n}$ and $p\text{-Bottom}_{EN}$ mean applying defective layers with keep probability $p$ to bottom layers, applying defective layers with keep probability $p$ to top layers, making whole channels defective with keep probability $p$, using the same defective mask in every channel with keep probability $p$, increasing channel number to $n$ times at bottom layers and ensemble five models with different defective masks of the same keep probability $p$ respectively.

| Architecture | FGSM$_8$ | FGSM$_{16}$ | FGSM$_{32}$ | PGD$_{16}$ | PGD$_{2,16}$ | PGD$_{32}$ | CW$_{40}$ | Acc |
|---|---|---|---|---|---|---|---|---|
| ResNet-18 | 25.53% | 17.47% | 8.56% | 3.32% | 3.26% | 1.18% | 0.00% | 95.33% |
| 0.7-Bottom | 46.12% | 23.30% | 10.48% | 33.90% | 29.04% | 19.33% | 2.66% | 94.03% |
| 0.5-Bottom | 57.05% | 31.01% | 11.07% | 57.52% | 51.70% | 40.37% | 14.61% | 93.39% |
| 0.3-Bottom | 72.67% | 48.17% | 15.20% | 82.46% | 77.49% | 71.30% | 39.89% | 91.83% |
| 0.25-Bottom | 78.23% | 58.19% | 21.20% | 87.86% | 84.06% | 77.91% | 47.33% | 91.46% |
| 0.2-Bottom | 82.27% | 61.61% | 19.70% | 91.00% | 87.92% | 83.72% | 51.14% | 91.18% |
| 0.15-Bottom | 85.80% | 65.92% | 22.73% | 94.00% | 91.93% | 88.82% | 57.36% | 90.15% |
| 0.1-Bottom | 92.93% | 79.13% | 48.34% | 96.49% | 95.94% | 94.39% | 65.63% | 87.68% |
| 0.05-Bottom | 94.77% | 87.13% | 63.36% | 97.74% | 97.26% | 95.82% | 69.14% | 84.53% |
| 0.7-Top | 23.76% | 16.66% | 9.52% | 2.39% | 2.37% | 0.84% | 0.00% | 95.16% |
| 0.5-Top | 23.01% | 12.19% | 6.56% | 3.35% | 3.27% | 1.18% | 0.00% | 94.94% |
| 0.3-Top | 22.87% | 11.61% | 6.63% | 5.22% | 4.84% | 1.95% | 0.19% | 94.61% |
| 0.5-Bottom, 0.5-Top | 47.85% | 18.29% | 12.01% | 48.59% | 42.03% | 31.72% | 15.84% | 92.48% |
| 0.7-Bottom, 0.7-Top | 42.34% | 22.07% | 7.84% | 27.19% | 23.31% | 14.06% | 1.70% | 94.16% |
| 0.7-Bottom, 0.3-Top | 31.43% | 12.17% | 6.34% | 19.14% | 15.40% | 8.60% | 1.53% | 93.44% |
| 0.3-Bottom, 0.3-Top | 57.26% | 29.36% | 13.99% | 71.03% | 63.90% | 55.09% | 30.00% | 89.78% |
| 0.3-Bottom, 0.7-Top | 68.66% | 41.32% | 13.72% | 78.61% | 74.22% | 66.02% | 35.71% | 91.23% |
| 0.5-Bottom$_{DC}$ | 30.81% | 14.77% | 6.08% | 6.18% | 5.38% | 2.33% | 0.00% | 95.12% |
| 0.3-Bottom$_{DC}$ | 34.57% | 17.32% | 8.04% | 10.68% | 9.58% | 4.86% | 0.19% | 94.23% |
| 0.1-Bottom$_{DC}$ | 43.46% | 17.61% | 10.54% | 39.53% | 34.10% | 25.01% | 7.41% | 93.27% |
| 0.5-Bottom$_{SM}$ | 71.27% | 49.21% | 16.27% | 80.23% | 74.38% | 65.92% | 34.92% | 92.57% |
| 0.3-Bottom$_{SM}$ | 79.48% | 49.66% | 15.65% | 90.74% | 87.33% | 82.03% | 48.28% | 89.81% |
| 0.1-Bottom$_{SM}$ | 65.85% | 42.59% | 21.87% | 80.36% | 75.71% | 71.25% | 43.22% | 74.28% |
| 0.5-Bottom$_{\times 2}$ | 44.13% | 21.71% | 9.49% | 32.98% | 27.34% | 17.64% | 2.65% | 94.12% |
| 0.3-Bottom$_{\times 2}$ | 60.51% | 30.89% | 11.58% | 66.09% | 59.08% | 48.06% | 21.52% | 93.01% |
| 0.1-Bottom$_{\times 2}$ | 85.26% | 67.91% | 32.51% | 92.46% | 89.99% | 85.86% | 52.96% | 90.49% |
| ResNet-18$_{EN}$ | 27.36% | 17.72% | 8.49% | 2.50% | 2.58% | 0.72% | 0.00% | 96.03% |
| 0.5-Bottom$_{\times 2,EN}$ | 48.07% | 20.83% | 10.35% | 37.11% | 31.01% | 19.68% | 4.91% | 95.10% |
| 0.5-Bottom$_{EN}$ | 60.42% | 31.08% | 10.77% | 60.63% | 54.00% | 41.55% | 13.61% | 94.56% |
| 0.3-Bottom$_{EN}$ | 76.08% | 51.49% | 13.19% | 85.51% | 80.85% | 73.29% | 39.51% | 93.31% |
| 0.1-Bottom$_{EN}$ | 94.40% | 81.32% | 48.52% | 97.58% | 96.87% | 95.33% | 66.99% | 89.45% |

Table 17: Extended experimental results of Section 4.3. Numbers in the middle mean the success defense rates. Adversarial examples are generated against *SENet-18*. The model trained on CIFAR-10 achieves 95.15% accuracy on test set. $p$-Bottom, $p$-Top, $p$-Bottom$_{DC}$, $p$-Bottom$_{SM}$, $p$-Bottom$_{\times n}$ and $p$-Bottom$_{EN}$ mean applying defective layers with keep probability $p$ to bottom layers, applying defective layers with keep probability $p$ to top layers, making whole channels defective with keep probability $p$, using the same defective mask in every channel with keep probability $p$, increasing channel number to $n$ times at bottom layers and ensemble five models with different defective masks of the same keep probability $p$ respectively.

| Architecture | FGSM$_8$ | FGSM$_{16}$ | FGSM$_{32}$ | PGD$_{16}$ | PGD$_{2,16}$ | PGD$_{32}$ | Acc |
|---|---|---|---|---|---|---|---|
| ResNet-18 | 37.67% | 20.25% | 5.40% | 26.97% | 20.65% | 17.58% | 95.33% |
| 0.7-Bottom | 50.06% | 23.54% | 9.53% | 45.27% | 36.74% | 31.61% | 94.03% |
| 0.5-Bottom | 57.35% | 30.52% | 11.13% | 58.66% | 50.82% | 43.89% | 93.39% |
| 0.3-Bottom | 71.75% | 47.35% | 15.47% | 77.57% | 72.68% | 64.06% | 91.83% |
| 0.25-Bottom | 76.81% | 56.69% | 19.44% | 83.32% | 79.23% | 70.72% | 91.46% |
| 0.2-Bottom | 79.46% | 61.45% | 21.36% | 87.55% | 84.41% | 76.35% | 91.18% |
| 0.15-Bottom | 85.51% | 66.55% | 25.35% | 91.65% | 89.29% | 82.90% | 90.15% |
| 0.1-Bottom | 92.58% | 80.68% | 51.90% | 93.41% | 92.83% | 90.30% | 87.68% |
| 0.05-Bottom | 95.24% | 87.10% | 64.22% | 93.14% | 92.78% | 91.14% | 84.53% |
| 0.7-Top | 36.72% | 18.97% | 9.65% | 26.91% | 20.87% | 17.37% | 95.16% |
| 0.5-Top | 35.93% | 13.80% | 2.99% | 26.36% | 21.31% | 17.35% | 94.94% |
| 0.3-Top | 34.05% | 13.06% | 4.04% | 29.84% | 23.08% | 19.04% | 94.61% |
| 0.5-Bottom, 0.5-Top | 50.78% | 19.25% | 9.12% | 52.42% | 45.66% | 38.49% | 92.48% |
| 0.7-Bottom, 0.7-Top | 47.36% | 22.74% | 5.04% | 41.42% | 34.26% | 28.90% | 94.16% |
| 0.7-Bottom, 0.3-Top | 40.38% | 13.50% | 3.28% | 36.96% | 29.80% | 24.98% | 93.44% |
| 0.3-Bottom, 0.3-Top | 59.19% | 28.00% | 12.13% | 68.60% | 62.34% | 53.50% | 89.78% |
| 0.3-Bottom, 0.7-Top | 67.14% | 40.57% | 13.80% | 73.93% | 69.07% | 60.58% | 91.23% |
| 0.5-Bottom$_{DC}$ | 37.37% | 16.99% | 6.62% | 26.39% | 20.09% | 16.79% | 95.12% |
| 0.3-Bottom$_{DC}$ | 42.39% | 19.90% | 6.74% | 28.85% | 22.83% | 20.22% | 94.23% |
| 0.1-Bottom$_{DC}$ | 47.41% | 21.12% | 11.43% | 45.20% | 38.32% | 32.68% | 93.27% |
| 0.5-Bottom$_{SM}$ | 69.61% | 46.57% | 14.85% | 76.26% | 70.62% | 61.99% | 92.57% |
| 0.3-Bottom$_{SM}$ | 79.69% | 48.86% | 13.87% | 87.03% | 83.43% | 76.01% | 89.81% |
| 0.1-Bottom$_{SM}$ | 67.77% | 44.38% | 20.74% | 68.67% | 66.66% | 61.88% | 74.28% |
| 0.5-Bottom$_{\times 2}$ | 46.93% | 21.74% | 7.11% | 45.08% | 37.12% | 31.21% | 94.12% |
| 0.3-Bottom$_{\times 2}$ | 60.23% | 29.72% | 11.07% | 63.62% | 57.28% | 48.83% | 93.01% |
| 0.1-Bottom$_{\times 2}$ | 83.32% | 66.44% | 33.11% | 89.57% | 87.33% | 80.20% | 90.49% |
| ResNet-18$_{EN}$ | 39.75% | 18.73% | 6.59% | 26.87% | 20.33% | 17.22% | 96.03% |
| 0.5-Bottom$_{\times 2,EN}$ | 51.91% | 19.60% | 7.86% | 49.29% | 39.53% | 34.29% | 95.10% |
| 0.5-Bottom$_{EN}$ | 60.43% | 31.07% | 10.50% | 61.60% | 53.91% | 46.50% | 94.56% |
| 0.3-Bottom$_{EN}$ | 74.11% | 50.89% | 13.54% | 80.75% | 76.27% | 66.71% | 93.31% |
| 0.1-Bottom$_{EN}$ | 94.14% | 82.46% | 52.59% | 96.22% | 95.20% | 92.64% | 89.45% |

Table 18: Extended experimental results of Section 4.3. Numbers in the middle mean the success defense rates. Adversarial examples are generated against *VGG-19*. The model trained on CIFAR-10 achieves 94.04% accuracy on test set. $p$-Bottom, $p$-Top, $p$-Bottom$_{DC}$, $p$-Bottom$_{SM}$, $p$-Bottom$_{\times n}$ and $p$-Bottom$_{EN}$ mean applying defective layers with keep probability $p$ to bottom layers, applying defective layers with keep probability $p$ to top layers, making whole channels defective with keep probability $p$, using the same defective mask in every channel with keep probability $p$, increasing channel number to $n$ times at bottom layers and ensemble five models with different defective masks of the same keep probability $p$ respectively.

