# OpenReview forum: "Defective Convolutional Networks"
_ICLR.cc/2021/Conference — Reject_

### Official Review · AnonReviewer1 · 2020-10-23
**circumstantial evidence. adversarial attacks not strong enough.**

**Rating:** 6
**Confidence:** 4

**Review:**

The authors present a modification of a convolutional layer that they claim forces their model to be more robust to texture attacks. The authors perform a variety of experiments attempting to show that non-texture features are being

The authors make a very strong claim in this work: that by deforming the output of convolution, they force the model to rely on other cues, such as shape. However, their analysis showing this to be the case is circumstantial. In none of their experiments do they explictly show that the model is focusing on shape outlines. For example, in the case of the random shuffling experiment, there might be a dependency on shape that is being obstructed, but there could also be a dependence on spatial relations. This is just one possibility.  It also doesn't make sense to use a convnet in the case of a 8x8 random shuffle; how do 1x1 convolutional layers on this experiment do? The Stylized-Imagenet experiment is potentially compromised by virtue of being very out of distribution to the ordinary ImageNet.

The authors should perhaps examine the visual attention literature to understand which pixels are being used to make classification decisions, such as Grad-CAM. Another set of experiments can involve using a dataset consisting solely of objects to be classified (ie, no background context), and which are ideally composed only of simple shapes. Would the model require training to be robust to such deformities.

Wrt the adversarial robustness experiments, it is not clear that their method yields more robust results or simply obfuscates the gradient attack so that a stronger attack is required. I would expect to see a plot showing their model's inference accuracy against attack strength, as measured by the number of PGD iterations. Their accuracy will decrease and should stabilize somewhere. Another possibility is to run gradient-free attacks, as in [1, 2]. We can then see whether their method yields more robust accuracies against an attack.

[1] Jonathan Uesato, Brendan O’Donoghue, Aaron van den Oord, and Pushmeet Kohli. Adversarial risk and the dangers of evaluating against weak attacks. In ICML, 2018.

[2] Anish Athalye, Nicholas Carlini, and David Wagner. Obfuscated gradients give a false sense of security: Circumventing defenses to adversarial examples. In ICML, 2018.


==================================================================

I thank the authors for their thorough comments and experimental details. I am more satisfied after reading them. I am raising my score from 4 to a 6.

---

> ### Author Response · Authors · 2020-11-20
> **Response to Reviewer 1 (Part 1)**
>
> We thank the reviewer for the valuable review and feedback. The experiments you suggest would help show the proposed method, and we can together make the paper better. During the rebuttal, we clarify some points to show the proposed evidence is not just circumstantial and the used adversarial attacks are strong. We would also articulate our claims to be better aligned with the evidence and appreciate it if you have some suggestions here. **If the new comment still can not fully address your concerns, please let us know as soon as possible.** Then, we can appropriately take the new feedback into account during the discussion phase.
>
> ***Q1***:  The current analysis is circumstantial and none of the experiments show that the model is focusing on shape outlines. Examine Grad-CAM. Also, certain concerns about the random shuffling experiment and the Stylized-Imagenet experiment.
>
> ***A1***:  How to prove the model makes predictions relying less on textural information but more on shape information? (The definition of shape and texture is according to [1]) We’ve been thinking a lot and proposed four empirical pieces of evidence in Sec. 3.2, including the random shuffling experiment, the stylized-imagenet experiment, the shape-changed adversarial examples, and the robustness gains against standard adversarial perturbations and additive Gaussian noise. Note that we have tested on 5 common architectures, including ResNet-18, ResNet-50, DenseNet-121, SENet-18, and VGG-19.
>
> First, we believe that the shape-changed adversarial examples can explicitly show the model is focusing on shape cues. Fig. 3,6,7 show the generated adversarial examples by the proposed model would change the global object shape, while the standard ones cannot. The adversarial examples are generated by modifying the original images based on the gradients, which indicates the place on original images pays attention to. For the Grad-CAM, it may not be suitable for use here, due to Grad-CAM would use a coarse heatmap of the same size as the convolutional feature maps and cannot tell the edge of objects.
>
> In the random shuffle experiments, the image patches are randomly relocated. The global object shape is destroyed while the local texture is preserved. We perform the experiment and find the proposed ones consistently perform worse than standard ones. So we use it as evidence to indicate the proposed ones may more rely on shape information for predicting. However, the spatial relations of those patches are totally deconstructed and the trained model has no information about the randomly relocated of the validation data. Therefore, we think that the hypothesis, there could be a dependence on spatial relations, cannot be established. During the rebuttal, we find that the random shuffling experiment is also used to examine if the model is biased towards shape in the literature [2,3].
>
> In the Stylized-ImageNet experiments, the image context and texture are replaced with other sources. The textural information is severely distorted while the global shape is preserved. In the experiment, the standard CNNs and the proposed ones are tested on the same test data. Therefore, the mentioned hypothesis cannot be established, since the two kinds of models face the same out-of-distribution situations but have significant performance differences (~6%). Also, the stylized-imagenet experiment is used in [1].
>
> To sum up, we have included evidence, the shape-changed adversarial examples, to explicitly show the proposed model is biased towards shape cues. Also, we discuss further the random-shuffle and stylized-imagenet experiments above and hope can address your concerns. **If you still believe they are very weak, we can appropriately ask the AC and other reviewers to join the discussion and reach a consensus.**
>
> [1] ImageNet-trained CNNs are biased towards texture; increasing shape bias improves accuracy and robustness, ICLR19
>
> [2] Interpreting Adversarially Trained Convolutional Neural Networks, ICML19
>
> [3] Deep convolutional networks do not classify based on global object shape, PLoS Computational Biology18
>
> [4] Robustness May Be at Odds with Accuracy, ICLR19

---

> > ### Author Response · Authors · 2020-11-20
> > **(Part2)**
> >
> > ***Q2***: Regarding adversarial attacks not strong enough and suggest running gradient-free attacks.
> >
> > ***A2***: In this paper, we’ve tested against FGSM, PGD, ensemble-PGD, CW, Boundary Attack, Gaussian additive noise. For the PGD, we’ve used step size $\in \\{1, 2, 4\\}$, step numbers $\in \\{2, 4, 5, 8, 10, 12, 20, 40\\}$, and perturbation scale $\in \\{4, 8, 16, 32\\}$. Please refer to Sec. 4 and Appendix A. If there are further requests, please let us know.
> >
> > Also, **we have performed a gradient-free attack**, Boundary Attack[1], which is an attack relying on the final model decision and has no access to the gradient. We mentioned it in the contribution section and Sec.4 as the decision-based attack. Please also refer to A.1.1 to see the experimental setting and results.
> >
> > [1] Decision-based adversarial attacks: Reliable attacks against black-box machine learning models, ICLR18
> >
> > ------------------------------------------------------
> >
> > ***Q3***: Regarding the obfuscated gradient.
> >
> > ***A3***: There may be some misunderstandings. The obfuscated gradients come from wrongly estimating gradients may due to the defense is non-differentiable resulting in numeric instability, contains randomization causing gradients to become randomized, or has multiple inner iterations causing gradients to explode [1]. However, they all do not exist in the proposed method. The proposed defense is fully differentiable, the structure is determined before training and testing, and no inner iteration exists. Therefore, **there is no chance to wrongly estimate the gradient and result in obfuscated gradients.** Please let us know if you still doubt this point.
> >
> > [1] Obfuscated Gradients Give a False Sense of Security: Circumventing Defenses to Adversarial Examples, ICML18
> >
> > ------------------------------------------------------
> > ***Q4***:  Another set of experiments can involve using a dataset consisting solely of objects to be classified (ie, no background context), and which are ideally composed only of simple shapes. Would the model require training to be robust to such deformities?
> >
> > ***A4***:  Thanks for the suggestion of the additional experiment. **We are not very sure about the details according to the current descriptions.** Are you suggesting using dataset A which contains simple shapes with background context and dataset B which contains only edges of shapes without any context, and we train models on dataset A and directly test on dataset B to compare the performance? If so, we think the stylized-imagenet experiment serves the same role. The test data of the stylized-imagenet experiment contains the edge of objects and extra context from other sources, and may exhibit a stronger test since the model needs not to be distracted by redundant context. If not, please describe it in more detail, then we can perform the experiment during the discussion or revision.

---

### Official Review · AnonReviewer4 · 2020-10-26
**A great paper that defends texture-related attack the with simple solution.**

**Rating:** 6
**Confidence:** 3

**Review:**

##########################################################################

Summary:

Studies suggest that CNNs that overly rely on texture features are more vulnerable to adversarial attacks. The authors of this paper propose a simple yet effective method "defective convolution" that randomly "disables" neurons on the convolution layer. The authors argue that by doing so, the CNN is encouraged to learn less from object texture and more on features such as shape. The authors support this statement by empirically evaluating the proposed model under multiple perturbation methods.

##########################################################################

Reasons for score:


Overall, I vote for accepting. This paper provides a simple yet novel method to force CNNs from learning texture-based features. But I found it more important to understand why such a simple method would achieve this effect rather than using it to defend against adversarial attacks. I hope the authors could provide more motivation and experiments to understand the effect that defective convolution layers have on the CNNs.


##########################################################################

Pros:

1.  This paper is addressing a very fundamental question of CNN: how can we change convolution filters so that the model will learn certain visual features (shape) while less likely to learn others (texture)?  To answer this question requires a better understanding of the underlying mechanism of CNNs. This work could serve as the initial step for answering this question.

2. The authors have conducted experiments on synthetically altered images to show that the defective convolution indeed tends to learn less from texture while more putting more emphasis on edges. The comparative experiments in section 4 also empirically support the author's statement that CNNs with such property is more robust against transfer-based attacks.

3. This paper is well-written. The introduction section provides sufficient background for the problem with clear intuition for the method the authors proposed. The literature review is sufficient and well-organized.


##########################################################################

Cons:

1. This paper needs better motivation. Would this work be applicable to other real-world CV application where the texture of the object of interest is a confounding factor?

2. The author argues in 3.1 that by using M_defect, neurons of the conv layers are masked out and therefore local features are hard to preserve. While this is intuitive, a more rigorous analysis would increase the credibility of this statement. What would be a proper mathematical definition of texture? How it is related to the locality? Why masking out spatial location in the conv layers would impact locality?


##########################################################################

Suggestions:

1. Please address the concerns in the cons section.

2. The ablation study mostly explores the effect of p and the layers where defective convolution is inserted. Intuitively, I think the spatial location where the defective neurons are placed would also impact the model's behavior. For instance, if all defective neurons are on the edge of the filter, we essentially reduce a large filter to a smaller one. On the other hand, if the defective neurons are at the center of the filters then we could discourage the model from learning some continuous patterns. It would be interesting to see how this would affect the model's performance.



#########################################################################

---

> ### Author Response · Authors · 2020-11-20
> **Response to Reviewer 4**
>
> We thank the reviewer for the insightful feedback and the recommendation. The comments are really profound, and we would like to share some of our thoughts below.
>
> ***Q1***: Would this work be applicable to other real-world CV applications where the texture of the object of interest is a confounding factor?
>
> ***A1***: Our humans can still recognize the photos containing defective pixels since we perceive objects by their shape while CNN does not tend to be [1,2]. This work is motivated by this point. Combining with the recent finds that the current adversarial attack is mainly based on distorted texture, our work naturally can defend the attack. Beyond this point, we find the adversarial examples generated by the proposed method change the original object shape and may fool humans (Fig. 3). This means the learned feature may be shape-aware and better aligned with human perception. Therefore, if there are related research or applications that need the learning model to be aligned with human perception, the proposed representation learning technique may help.
>
> [1] ImageNet-trained CNNs are biased towards texture; increasing shape bias improves accuracy and robustness, ICLR19
>
> [2] Deep convolutional networks do not classify based on global object shape, PLoS Computational Biology18
>
> -------------------------------
>
> ***Q2***: What would be a proper mathematical definition of texture? How is it related to the locality? Why masking out spatial location in the conv layers would impact locality?
>
> ***A2***: Texture is characterized by the spatial distribution of gray levels in a neighborhood[1]. Consider a local piece of an image containing a certain texture, the response will be strong if we apply a specific convolution kernel. For a CNN and a dataset, a filter can learn to have a strong response with certain texture since it repeats and the model would use it to recognize the class. This is also the story that goes on [2], where a cat with elephant skin would be classified as an elephant. Here, we mask out certain neurons of filters in the bottom layer of a CNN, where those filters originally respond to detect texture [3]. For a masked filter (3x3), the types of texture within a neighbor (i.e. locality) it can detect are limited. If we severely limit the texture it can detect, it has to seek another signal that can be responded via a multi-layer combination to classify. Sorry for we can only give an intuitive explanation at present. Do you think we should add the above intuitive analysis in the main paper? Also, if you have further suggestions on the proper mathematical modeling, we are very glad to hear.
>
> [1] Machine vision, McGraw-hill New York1995
>
> [2] ImageNet-trained CNNs are biased towards texture; increasing shape bias improves accuracy and robustness, ICLR19
>
> [3] Visualizing and understanding convolutional networks, ECCV2014
>
> -------------------------------
>
>
> ***Q3***: Intuitively, I think the spatial location where the defective neurons are placed would also impact the model's behavior. It would be interesting to see how this would affect the model's performance. Consider all defective neurons are on the edge or on the center of the filter.
>
> ***A3***: The mentioned point is very insightful and is also what we plan to study in the next step. When it comes to the spatial location, we consider the general spatial location of a feature map [w, h, c], where w, h, c denote width, height, and channel. In the current ablation study, we compare with the defective channel and the shared mask, where both two cases can be viewed as the special cases of the proposed method. For the defective channel, we apply the defective neurons in a whole channel way and find there are no improvements on robustness. For the shared mask, we apply the defective neurons to share across different channels and find this would result in significant performance dropping. The two variants mainly study the channel dimension (c of [w,h,c]).
>
> However, for the [w,h], it is not easy to set some meaningful special cases. Note that, the filter would go over the whole feature map, thus we cannot set all defective neurons on the edge or on the center of the filter. Will changes of [w,h] distributions result in different inductive biases? The answer is Yes. Consider the results of gray-box attacks in A.6, the proposed ones show superior performance than the standard one. This indicates that defective CNNs with different distributions of defective neurons would yield larger differences in learned representation than the standard ones. Therefore, we suggest studying [w,h] with the recent advances in NAS in the future, where we can search the distribution of defective neurons on [w,h] for downstream tasks.

---

### Official Review · AnonReviewer3 · 2020-10-27
**An ok paper**

**Rating:** 6
**Confidence:** 4

**Review:**

Summary: The paper proposes a method to improve adversarial robustness of the current convolutional networks. The method is based on dropping outputs of a fraction of neurons. However, unlike in dropout the masks are kept fixed throughout training/inference and applied to the bottom layers of the network. This shifts the focus of the network away from texture and towards shape features resulting in the adversarial examples being fooling humans as well.

I am inclined to vote for accepting the paper. The approach is novel as it tackles the issue of adversarial examples from a different angle than the usual denoising appraches, however, there is some room for improvement.

Possible improvements and questions:
- EGC-FL (Yuan&He'20) performs equally well in the experiments, the authors argue that the main advantage of their method is the lower runtime but do not give any quantitative information on runtime. This information needs to be presented/commented upon in more detail.
- writing/clarity can be improved in the experiments section, sometimes the claims seem a bit exaggerated (e.g., p8 saying "This corroborates the phenomena..." when there was no result showing the said phenomena)
- the authors claim Defective CNNs can help saving a lot of computate and storage, is this hypothetical or is this a real option with current deep learning libraries?
- why in some tables (2&3) keep probability is 0.1 while in others (4) it is 0.3 & 0.5? Without explanation this seems like cherry picking.
- from the experiment with shared defective masks it seems that the Defective CNNs actually encode the texture as well, the information is just spread across channels, or can the result be interpreted otherwise?

Small corrections:
- Eqs 1,3,7 could use more standard notation for convolution to improve readability ($*$ instead of the current $\bigotimes_{conv}$)
- Eq 6 could spell out $M_{dp}$ as $M_{dropout}$ to be consistent with $M_{defect}$ in Eq 5
- Sec 3.2: "by both two CNNs" -> "by both of the CNNs"
- p5: "... and Gaussian noises usually hard to affect" -> "...and Gaussian noise usually hardly affect"
- Sec 4.1.1: "approaches try to erase" -> "approaches that try to erase"
- Sec 4.2: "the chose of" -> "the choice of" and "Without loss of generality" -> "Here" and "v.s" -> "vs."

---

> ### Author Response · Authors · 2020-11-20
> **Response to Reviewer 3**
>
> We thank the reviewer for the detailed feedback and the positive rating. We believe we can address your concerns below and would incorporate updates in the revised version.
>
> ***Q1***: EGC-FL (Yuan&He'20) performs equally well in the experiments, the authors argue that the main advantage of their method is the lower runtime but do not give any quantitative information on runtime. This information needs to be presented/commented upon in more detail.
>
> ***A1***: Thanks for mentioning this point. According to the original paper [1], we find their method would cost extra time to collect adversarial examples (e.g. generated adversarial examples with 8 step size would need 8 times extra time), and run 4 times inner loops that the inference time is 4 times that of our method. Since they do not release code as we will do, we cannot compare the specific runtime with them at this time.
>
> [1] Ensemble Generative Cleaning with Feedback Loops for Defending Adversarial Attacks, ECCV20
>
> --------------------------------------------------
> ***Q2***: Sometimes the claims seem a bit exaggerated, e.g. p8 saying "This corroborates the phenomena..." when there was no result showing the said phenomena.
>
> ***A2***:  For the specific case, we actually have the phenomenon that applying defective layers to bottom layers can achieve better performance while the top layers cannot. We want to argue this finding is consistent with our tuition and also the literature. We’ve rewritten this paragraph now. If you further find other improper statements, please let us know, then we will articulate our claims precisely.
>
> --------------------------------------------------
>
>
> ***Q3***: Defective CNNs can help to save a lot of computing and storage, is this hypothetical or is this a real option with current deep learning libraries?
>
> ***A3***: We can achieve it with the index operations used in [Sparse ConvNet](https://github.com/facebookresearch/SparseConvNet) which is implemented based on current deep learning libraries. The index operations can help avoid computing the value of the deactivated neurons no matter what the input is, thus help save the computation and storage cost. Moreover, the proposed principle may be naturally integrated into FPGA for edge-computing.
>
> --------------------------------------------------
>
>
> ***Q4***: Why in some tables (2&3) the probability is 0.1 while in others (4) it is 0.3 & 0.5?
>
> ***A4***: We want to note that it is enough to show our improvements with either 0.5 or 0.3 in Table 4. The reason why we show the results of both 0.3 and 0.5 is to show the trend of robustness increases as the keep probability becomes smaller. Since there is a trade-off between the accuracy and robustness (discussed in Sec. 4.2 and A.9), we use 0.1 to compare with the SOTA methods. We would make this clear in the revision. Second, we’ve tested the structure with keep probabilities ranging from 0.05 to 0.7 and listed the results in Table 14, 15, 16, 17, 18.
>
> --------------------------------------------------
>
>
> ***Q5***: With shared defective masks it seems that the Defective CNNs actually encode the texture as well, the information is just spread across channels, or can the result be interpreted otherwise?
>
> ***A5***: The Defective CNN with shared defective masks would also be biased towards the shape. You can see the improvements in the robustness in Table 5. But, such monotone patterns across channels would cause the accuracy to significantly drop.
>
> --------------------------------------------------
>
>
>
> Thanks for the detailed corrections, we've fixed them in revision.

---

> > ### Comment · AnonReviewer3 · 2020-11-23
> > **Response to the authors**
> >
> > Thank you for the answers. Can you please further clarify the following two points?
> >
> > - Ad index operations, I suppose you refer to the feature & pointer matrices in https://arxiv.org/pdf/1409.6070.pdf, correct? If so would this help to save a substantial compute? Sparse ConvNets work well with sparse inputs, but the sparsity in Defective CNNs varies between different filters. The probability of any neuron not being used in the next layer becomes negligible with increasing number of filters.
> >
> > - Ad varying keep probability in tables, I understand what you mean, but the change seems to be ad-hoc and as a reader one starts to wonder. If you already have all the numbers, than I would propose to change the p=0.3 to p=0.1 in Table 4. Connected to this, should not the number 15.96% (Standard CNN, PGD in Table 2) appear somewhere in Table 4?

---

> > > ### Author Response · Authors · 2020-11-23
> > > **Response to Reviewer 3**
> > >
> > >
> > > Thank you very much for your timely update. We clarify the mentioned points as below. If there’re further assessments, please let us know, then we can appropriately take the new feedback into account.
> > >
> > > ---------------------------------------------
> > > ***Q1***: Regarding the index operations.
> > >
> > > ***A1***: Yes, we refer to the feature & pointer matrices in [1] and the hash-table techniques in [2], for efficiently computing only at active sites. Applying the defective neurons to feature maps actually causes the feature maps to sparse while not affect the convolutional filters. Therefore, the mentioned techniques would naturally support the proposed scenery. The convolutional filters of a defective CNN would still be the same as the standard ones but can be equivalently viewed that have different structures at different spatial locations of a feature map.
> > >
> > > [1] Spatially-sparse convolutional neural networks, 2014
> > >
> > > [2] Submanifold Sparse Convolutional Networks, CVPR18
> > >
> > > ------------------------------------------------
> > > ***Q2***: Regarding the varying keep probabilities in Tables.
> > >
> > > ***A2***: Thank you for mentioning this again. We got your point now that the inconsistent hyper-parameters seem to be ad-hoc for new readers. We are unaware of this point before and will follow your suggestion. Some results of p=0.1 are currently listed in Table 14-18 but not all, thus we would change this in the revision.
> > >
> > > The number 15.96% should not appear in Table 4, due to the experimental settings are different in Table 2 and Table 4. In Table 2, we aim to compare with SOTA methods and run PGD with perturbation scale $\ell_\infty=8/255$, steps $7$, and step size $2/255$. In Table 4, we run PGD with perturbation scale $\ell_\infty=16/255$, steps $20$, and step size $1/255$. The reason why has different values is that we also study how different perturbation scales, steps, and step sizes would affect the robustness in this paper. For PGD, we’ve conducted step size $\in \\{1, 2, 4\\}$, step numbers $\in \\{2, 4, 5, 8, 10, 12, 20, 40\\}$, and perturbation scale $\ell_\infty \in \\{4, 8, 16, 32\\}$. Those values are reasonable to be scaled up and compared, and we list the results of a set of typical parameters in Table 4. Parts of Table 4 numbers appear in Table 14-18.

---

### Decision · Program_Chairs · 2021-01-07
**Final Decision**

**Decision:**

Reject

**Comment:**

The paper presents a method to make CNN focus more on structure rather than texture by constraining a random set of neurons per feature map to have constant activation.
The paper has limited novelty and unclear analysis of the experimental results, for instance plots of accuracy vs strength of adversarial perturbation should be produced. Tables are not readable and results tend to be cluttered and confusing.  Some comparisons seem to be cherry picked as pointed out by some reviewers.
Although the approach seems to be well received by the reviewers they all shared similar concerns about having a stronger motivation and better validation of the approach (that is not amount of comparisons but the right comparisons that would clear doubts and make the work directly comparable to others).
I strongly encourage the authors to perform a deeper analysis and to clearly work on hypothesis and validation of their work. In my opinion, although the reviewers think different, the experiments are not sufficient to validate the strong claim of the paper.